# Fast Inference with Kronecker-Sparse Matrices

**Antoine Gonon** [* 1 2]  **Léon Zheng** [* 1 3 4]  **Pascal Carrivain** [* 1]  **Quoc-Tung Le** [1 5]

## Abstract

Kronecker-sparse (KS) matrices—whose supports are Kronecker products of identity and all-ones blocks—underpin the structure of Butterfly and Monarch matrices and offer the promise of more efficient models. However, existing GPU kernels for KS matrix multiplication suffer from high data movement costs, with up to $50\%$ of time spent on memory-bound tensor permutations. We propose a fused, output-stationary GPU kernel that eliminates these overheads, reducing global memory traffic threefold. Across 600 KS patterns, our kernel achieves in FP32 a median speedup of $\times 1.4$ and lowers energy consumption by $15\%$. A simple heuristic based on KS pattern parameters predicts when our method outperforms existing ones. We release all code[1] at github.com/PascalCarrivain/ksmm, including a PyTorch-compatible KSLinear layer, and demonstrate in FP32 end-to-end latency reductions of up to $22\%$ in ViT-S/16 and $16\%$ in GPT-2 medium.

## 1. Introduction

Matrix multiplications dominate the *time* and *energy* budgets of modern-day networks involving large linear layers. For instance, a single forward of a Vision Transformer (ViT-H/14) spends $\sim 60\%$ of its runtime in linear layers (Appendix E.7). Using structured sparse factors in place of dense weights—either learned from scratch or introduced after training—offers a compelling way to reduce memory usage and inference FLOPs (Dao et al., 2019; 2020;

2022a;b). Among the many structures explored, the *Butterfly/Monarch family* has gained traction in both vision and language models (Dao et al., 2019; 2022b; Fu et al., 2023). Its building blocks are matrices whose *support* is a Kronecker product of identities and all-ones blocks. We call them **Kronecker-sparse (KS) matrices** (Definition 2.1), and denote by $(a, b, c, d)$ the pattern of a KS matrix $\mathbf{K}$ with support $\mathbf{I}_a \otimes \mathbf{1}_{b \times c} \otimes \mathbf{I}_d$, where $\otimes$ is the Kronecker product, $\mathbf{I}_n$ is the $n \times n$ identity matrix, and $\mathbf{1}_{b \times c}$ is matrix of size $b \times c$ full of ones. Factorizing a dense matrix $\mathbf{W}$ into KS matrices $\mathbf{K}_1 \cdots \mathbf{K}_L$ can reduce inference cost, as the sequential matrix-vector multiplications $\mathbf{x} \mapsto \mathbf{K}_1 \cdots \mathbf{K}_L \mathbf{x}$ can be more efficient than computing the dense multiplication $\mathbf{x} \mapsto \mathbf{W} \mathbf{x}$. The canonical example is the fast Fourier transform (FFT): the $N \times N$ discrete Fourier transform (DFT) matrix can be factored into $L = \log_2(N)$ KS factors of size $N \times N$, with at most $\mathcal{O}(N)$ nonzero entries per factor (Dao et al., 2019). This reduces the overall cost of computing the DFT from $\mathcal{O}(N^2)$ (via naive multiplication with the original matrix $\mathbf{W}$) to $\mathcal{O}(N \log_2 N)$ (through sequential multiplication with the KS factors $\mathbf{K}_\ell$); see Figure 1.

*Figure 1.* **FFT as a fast Kronecker-sparse transform.** The dense DFT matrix $\mathbf{W}$ can be factored as $\mathbf{W} = \mathbf{K}_1 \cdots \mathbf{K}_L$ (up to a column permutation), where each KS factor $\mathbf{K}_\ell$ has support $\mathbf{I}_{2^{\ell-1}} \otimes \mathbf{1}_{2 \times 2} \otimes \mathbf{I}_{2^{L-\ell}}$. This factorization underlies the $\mathcal{O}(N \log N)$ efficiency of the FFT algorithm.

**Scope of This Work: Focus on Inference.** This work focuses on efficiently computing $\mathbf{Y} = \mathbf{X}\mathbf{K}^\top$, where $\mathbf{K} \in \mathbb{R}^{M \times N}$ is a fixed Kronecker-sparse matrix and $\mathbf{X} \in \mathbb{R}^{B \times N}$ is a dense matrix representing a batch of inputs. We assume $\mathbf{K}$ can be fully preprocessed offline, but inputs and outputs must follow a fixed, dense layout—PyTorch's default in our case—which cannot be altered. This reflects standard inference scenarios, where models are already trained and deployed, and data arrives in a prescribed format. Optimizing this phase is essential, as inference accounts for over $90\%$ of large-scale ML costs (HPCwire, 2019; Barr, 2019).

---

*Equal contribution [1]ENS de Lyon, CNRS, Inria, Université Claude Bernard Lyon 1, LIP, UMR 5668, 69342, Lyon cedex 07, France [2]Institute of Mathematics, EPFL, Lausanne, Switzerland [3]valeo.ai, Paris, France [4]Huawei Lagrange Mathematics and Computing Research Center, Paris, France [5]Toulouse School of Economics, Toulouse, France. Correspondence to: Antoine Gonon .

*Proceedings of the $42^{nd}$ International Conference on Machine Learning*, Vancouver, Canada. PMLR 267, 2025. Copyright 2025 by the author(s).

[1]Also integrated into the signal-processing oriented package lazylinop (Appendix D.1).

**Cost of Memory Operations.** The fastest current GPU implementations for Kronecker-sparse multiplication—namely, Monarch's implementation (Dao et al., 2022b), which we refer to as BMM (as it relies on batched general matrix multiplication), and the block-sparse format method (BSR)—follow a shared strategy: (i) permute the input to ensure that specific elements are stored contiguously in memory; (ii) invoke a high-performance multiplication routine, such as a general matrix multiplication (GEMM); and (iii) apply an inverse permutation to restore the output to its original layout.

However, our profiling (in Section 3 below) reveals that these two permutations—before and after the multiplication—along with the associated memory reads and writes, can account for up to 50 % of the total runtime (Figure 4). Notably, we find that the overhead from these data movement operations increases with the ratio $h(b,c) = \frac{b+c}{bc}$, a simple expression that we motivate analytically and validate empirically across 600 KS patterns in this paper.

**Contributions.** We show that the data movement costs caused by the two permutations are not unavoidable. By introducing a new *output-stationary* tiling scheme for KS matrix multiplication, we fuse the three GPU kernels—permute, GEMM, and inverse permute—into a *single* kernel, eliminating two global memory passes. Our main contributions are as follows:

(i) **Time & Energy Benchmark.** We introduce the first large-scale public benchmark of KS matrix multiplication on GPU, spanning 6 orders of magnitude in matrix sizes, several sparsity levels, and supporting both single (FP32) and half (FP16) floating-point formats. Our results confirm that the proportion of the runtime spent at memory rewritings grows with $h(b,c) = \frac{b+c}{bc}$.

(ii) **Single-Kernel Implementation.** We propose an *output-stationary* tiling that reduces the cost of memory operations. We release publicly our kernel implementation in CUDA *and* OpenCL, plus a drop-in `KSLinear` layer for PyTorch. Our kernel achieves a median speedup of ×1.4, and a median energy reduction of 15 % in FP32.

(iii) **Design Heuristic.** For a KS pattern $(a,b,c,d)$, we empirically show that the ratio $h(b,c) = \frac{b+c}{bc}$ predicts latency, while $d \times h(b,c)$ predicts energy, giving practitioners a one-line rule for pattern selection.

(iv) **End-to-End Impact.** We demonstrate that injecting KS sparsity in linear layers of Transformers such as ViT-S/16 in vision and GPT-2 medium in language cuts wall-clock inference in FP32 by 22 % and 16 % respectively, compared to the original dense implementation.

**Outline.** Section 2 formalizes KS matrices and reviews existing GPU algorithms on PyTorch. Section 3 quantifies the time they spend on memory permutations. Section 4 presents the fused kernel and its analysis. Section 5 benchmarks time and energy. Section 6 demonstrates network-level gains. Section 7 concludes with open directions.

## 2. Kronecker-Sparse Matrices in a Nutshell

A *Kronecker-sparse (KS)* matrix imposes constraints on the *locations* of its nonzero entries (i.e., its support), but places no restrictions on the values of those entries. The support is defined by a Kronecker product of three simple matrices—identity, all-ones, and identity—which forms the common structural core of the Butterfly, Monarch, Kaleidoscope, and related layers (Dao et al., 2019; 2022a;b; Lin et al., 2021; Fu et al., 2023).

**Definition 2.1** (KS pattern and matrix). A **KS pattern** is an integer 4-tuple $\boldsymbol{\pi} = (a,b,c,d)$. Its corresponding support is $\mathbf{S}_{\boldsymbol{\pi}} = \mathbf{I}_a \otimes \mathbf{1}_{b \times c} \otimes \mathbf{I}_d$ (see Figure 2). A matrix $\mathbf{K} \in \mathbb{R}^{abd \times acd}$ is $\boldsymbol{\pi}$-*Kronecker-sparse* if $\mathrm{supp}(\mathbf{K}) \subseteq \mathrm{supp}(\mathbf{S}_{\boldsymbol{\pi}})$, where $\mathrm{supp}(\cdot)$ is the set of matrix indices corresponding to nonzero entries. We write $\mathbf{K} \in \mathcal{K}_{\boldsymbol{\pi}}$.

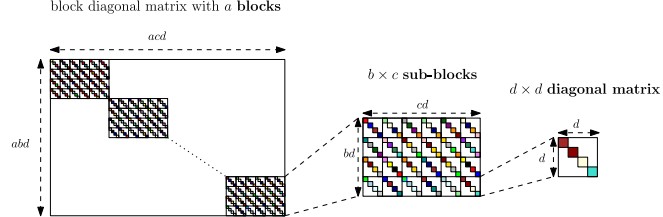

*Figure 2.* A $\boldsymbol{\pi}$-Kronecker-sparse matrix with $\boldsymbol{\pi} = (a,b,c,d)$ is a block-diagonal matrix with $a$ blocks, where each block itself is a block matrix composed by $b \times c$ diagonal matrices of size $d \times d$. The colored cells correspond to the nonzeros. We color the cells with different colors to indicate that the corresponding weights are free to take different values.

The mask $\mathbf{S}_{\boldsymbol{\pi}}$ has shape $abd \times acd$ and $abcd$ nonzeros, so the density (of nonzero entries) is $1/(ad)$. Increasing either $a$ or $d$ makes the factor sparser. See Appendix B for:

- a discussion on where KS fits in the wider sparsity landscape, with in particular Table 4 showing how popular structured layers in neural networks map to concrete KS patterns;
- links between KS matrix multiplication, sparse 3D convolutions, and sparse-tensor compilers;
- a brief overview of techniques with provable guarantees for approximating a dense matrix $\mathbf{W}$ by a KS product $\mathbf{K}_1 \cdots \mathbf{K}_L$. Note that this approximation problem lies outside the scope of this paper—we assume a fixed KS matrix $\mathbf{K}$ is given and focus on efficient multiplication with dense input batches $\mathbf{X}$; see Section 1.

## 2.1. Previous PyTorch GPU Implementations for KS

The fastest publicly available PyTorch implementations optimized for the KS structure follow the three-step procedure described in Algorithm 1: **(1)** permute the inputs; **(2)** apply a high-performance GEMM routine to $\tilde{\mathbf{K}}$, a pre-permuted, block-diagonal version of $\mathbf{K}$; **(3)** permute the output back to its original layout.

---

**Algorithm 1** Permutation-based KS matmul

---

**Require:** KS pattern $\boldsymbol{\pi} = (a, b, c, d)$, inputs $\mathbf{X} \in \mathbb{R}^{B \times acd}$,
  permutation matrices $\mathbf{P}, \mathbf{Q}$ (as in Appendix C.1)
  pre-permuted KS matrix $\tilde{\mathbf{K}} := \mathbf{P}^\top \mathbf{K} \mathbf{Q}^\top$
**Ensure:** outputs $\mathbf{Y} := \mathbf{X} \mathbf{K}^\top \in \mathbb{R}^{B \times abd}$
 1: $\tilde{\mathbf{X}} \leftarrow \mathbf{X} \mathbf{Q}^\top$ {permute columns}
 2: $\tilde{\mathbf{Y}} \leftarrow \tilde{\mathbf{X}} \tilde{\mathbf{K}}^\top$ {matrix multiply}
 3: $\mathbf{Y} \leftarrow \tilde{\mathbf{Y}} \mathbf{P}^\top$ {permute columns back}

---

**Any pair of permutation matrices $\mathbf{P}$ and $\mathbf{Q}$ in Algorithm 1 yields the correct KS matrix multiplication result.** Indeed, any permutation matrices $\mathbf{P}$ and $\mathbf{Q}$ satisfy $\mathbf{P}\mathbf{P}^\top = \mathbf{Q}^\top\mathbf{Q} = \mathbf{I}$, so this ensures that the multiplication of a batch of inputs $\mathbf{X} \in \mathbb{R}^{B \times acd}$ with a KS factor $\mathbf{K} \in \mathbb{R}^{abd \times acd}$ is equal to:

$$\mathbf{Y} = \mathbf{X}\mathbf{K}^\top = \underbrace{\mathbf{X}\mathbf{Q}^\top}_{:=\tilde{\mathbf{X}}} \underbrace{\mathbf{Q}\mathbf{K}^\top\mathbf{P}}_{:=\tilde{\mathbf{K}}^\top} \mathbf{P}^\top = \tilde{\mathbf{X}}\tilde{\mathbf{K}}^\top\mathbf{P}^\top.$$

**Existing implementations use a *specific* pair of permutations $(\mathbf{P}, \mathbf{Q})$ to achieve high efficiency.** These are chosen so that the permuted matrix $\tilde{\mathbf{K}} := \mathbf{P}^\top \mathbf{K} \mathbf{Q}^\top$ becomes **block-diagonal** with dense sub-blocks (details in Appendix C.1), enabling fast parallel multiplication in Step (2) using standard routines on each dense sub-block.

The main differences between existing implementations of Algorithm 1 lie in how they perform Step (2): either via batched GEMM (BMM) (Dao et al., 2022b) or block-sparse multiplication (BSR, for Block Compressed Sparse Row); see Table 1. We will also compare these to a direct tensor contraction baseline tailored to KS structure—EINSUM—inspired by Dao et al. (2022b). Implementation details are provided in Appendix A.

*Table 1.* Two PyTorch realizations of Algorithm 1.

|  | BMM | BSR |
|---|---|---|
| $\tilde{\mathbf{K}}$ storage | $(ad, b, c)$ tensor | $(abd, acd)$ in BSR |
| Permute $\mathbf{Q}$ (line 1) | torch.reshape | |
| Multiply step (line 2) | torch.bmm | linear |
| Permute $\mathbf{P}$ (line 3) | torch.reshape | |

The convenience of reducing the problem to a matrix multiplication with the block-diagonal matrix $\tilde{\mathbf{K}}$ comes at a cost: it requires two full-tensor permutations in Steps (1) and (3), which we find to consume a non-negligible portion of the runtime (Section 3). To address this, we introduce a new *mathematically equivalent* algorithm that avoids these permutations entirely (Section 4).

## 2.2. Reminder on Memory-Layout

PyTorch uses row-major storage for tensors. Placing the batch dimension first (*batch-size-first*[2]) ensures that each sample is stored contiguously in memory; placing it last (*batch-size-last*) instead makes the features contiguous. The historical default in most machine learning pipelines (including PyTorch) is *batch-size-first*. While the primary focus of this paper is to compare kernel implementations, we also study the effect of memory layout and find that *batch-size-last* performs better for KS matrices. To support both conventions, we provide a PyTorch `KSLinear` layer compatible with both layouts.

## 3. Cost of the Permutations in Baseline Implementations

As we show in Section 5, BMM is the fastest among existing implementations of KS matrix multiplication. It implements Algorithm 1 directly, with two explicit permutations—lines 1 and 3—that move data between global memory and registers before and after the multiplication with the block-diagonal matrix $\tilde{\mathbf{K}}$. Figure 3 (left) illustrates the data flow in this baseline.

In this section, we empirically demonstrate that these two permutations can account for up to 50 % of the total runtime, motivating the design of a new tiling strategy that avoids this overhead (Section 4).

### 3.1. Why Data Transfers Matter

GPU memory management plays a critical role in performance optimization. Memory on a GPU is organized hierarchically: global memory is large but slow, registers are small but fast, and shared memory sits in between (NVIDIA, 2024, Section 2.3). By default, data resides in global memory. When a kernel is executed, each GPU thread typically loads data from global memory into registers, performs register-level computations, and writes results back to global memory (Figure 3). Because memory access often becomes a performance bottleneck, minimizing data movement between global memory, shared memory, and registers is essential for an efficient implementation (NVIDIA, 2024, Section 5.3).

---

[2]We name the two memory layouts *batch-size-first* and *batch-size-last*, by analogy with PyTorch's *channels-last* optimization, which moves the channels dimension to the end for convolutions.

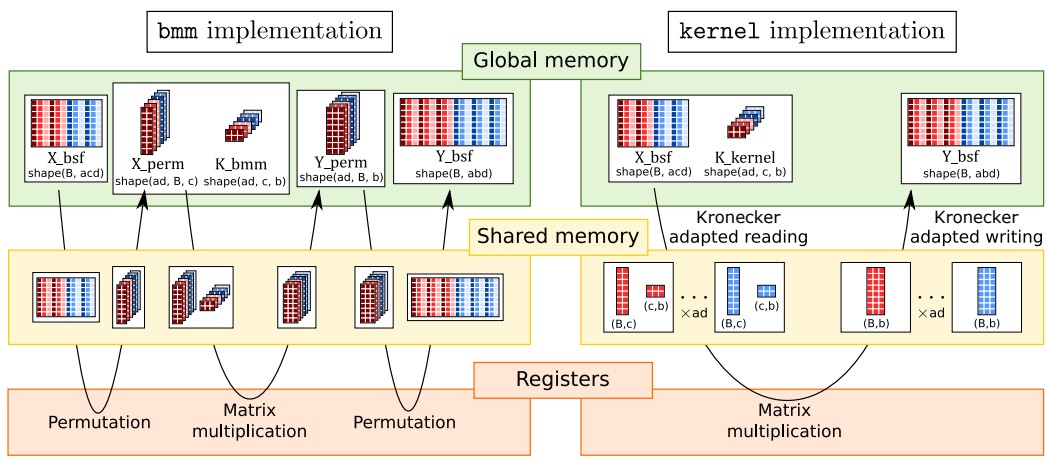

*Figure 3.* Data flow in BMM (three global passes) ([Dao et al., 2022b](#)) versus our fused KERNEL (one pass).

## 3.2. Cost of Data Transfers in Baseline Implementations

We focus on BMM as we will find out it is the fastest existing baseline. To isolate the cost of the memory rewriting operations corresponding to the permutations in Algorithm 1 (lines 1 and 3), we compare the runtime of BMM with a modified version in which these permutations are simply removed. Figure 4 ($y$-axis) shows that across 600 KS patterns, permutations consume up to 45 % of the time[3], and the share grows with $h(b, c) = \frac{b+c}{bc}$; we provide more explanation about this ratio in Section 4.

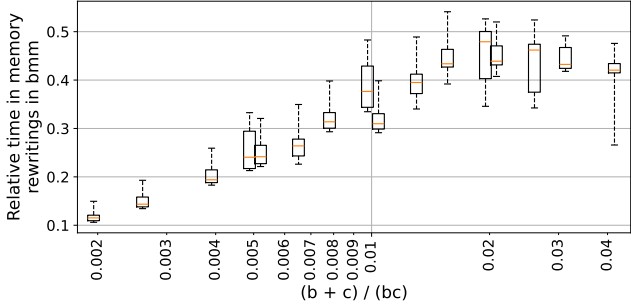

*Figure 4.* Share of runtime spent *only* on permutations in BMM (FP32, BSF layout). Each box aggregates patterns with the same $h(b, c) = \frac{b+c}{bc}$.

**Take-Away:** *Reducing the cost of memory transfers is a promising direction for improving the performance of Kronecker-sparse matrix multiplication. This is the main objective of our new implementation, described in the following section.*

---

[3]Regardless of the memory layout convention, *batch-size-first* or *batch-size-last* as shown in Appendix E.3.

## 4. Novel Tiling for KS Multiplication with Reduced Memory Transfers

Baseline implementations call three separate GPU kernels (PERMUTE-GEMM-PERMUTE); every call round-trips through global memory (Figure 3, left).

A naive approach to prevent such back and forths would be to directly try to merge these three different kernels into a single one. However, this is not as straightforward as it seems since a thread can only access data processed by other threads in the *same block* (no communication/synchronization across thread blocks). Thus, if a thread is assigned tiles (submatrices) on which to apply the three operations PERMUTE-MULTIPLY-PERMUTE, its workload cannot depend on the results obtained by a thread in *another* block. This requires rearranging the tiles assigned to each thread to ensure both *efficiency* and that no thread waits for the result of another thread in *another* block.

To this end, we introduce in Section 4.1 a novel, mathematically equivalent reformulation of the permutation-based algorithm (Algorithm 1), formalized in Algorithm 2. This reformulation leads to a new tiling strategy that enables us to implement the multiplication in a single GPU kernel, as described in Section 4.2. We then perform an analytical comparison of memory operations between our implementation and existing baselines (Section 4.3), and derive a heuristic that predicts when our kernel is expected to reduce the cost of the permutation-related memory operations of Algorithm 1. We validate this prediction empirically in Section 5.

### 4.1. New Tiling Strategy

We introduce a new tiling strategy for KS matrix multiplication, to reduce input/output (I/O) transfers between the different memory levels. Tiling consists of partitioning the

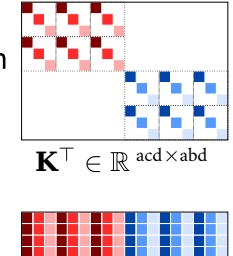

Initial sparse problem

Dense matrix multiplications in parallel

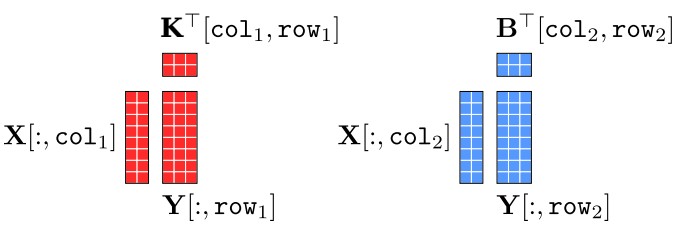

*Figure 5.* Illustration of Algorithm 2 for sparsity pattern $\boldsymbol{\pi} = (2, 3, 2, 3)$ and batch-size $B = 8$. The rows and columns $(\texttt{row}_1, \texttt{col}_1)$ are associated with $(i, j) = (0, 1)$ in the "for" loop of Algorithm 2, whereas $(\texttt{row}_2, \texttt{col}_2)$ is associated with $(i, j) = (1, 1)$.

matrices into smaller submatrices called *tiles*, which are processed in parallel, with the final result obtained by accumulating intermediate results from each tile (NVIDIA, 2024; 2023a). Our strategy arises from a *mathematically equivalent* reformulation of Algorithm 1, formalized as Algorithm 2, which we now explain.

---

**Algorithm 2** New mathematically equivalent tiling of Algorithm 1 (no global memory permutations), see Figure 5.

---

**Require:** pattern $\boldsymbol{\pi} = (a, b, c, d)$, inputs $\mathbf{X} \in \mathbb{R}^{B \times acd}$,
   KS factor $\mathbf{K} \in \mathcal{K}_{\boldsymbol{\pi}}$
**Ensure:** output $\mathbf{Y} := \mathbf{X}\mathbf{K}^\top \in \mathbb{R}^{B \times abd}$
1: $\mathbf{Y} \leftarrow \mathbf{0}$ {in global memory}
2: **for** $i = 0$ to $a - 1$, $j = 0$ to $d - 1$ **in parallel do**
3:    $\texttt{row} \leftarrow \{i\frac{M}{a} + j + kd : 0 \leq k < b\}$
4:    $\texttt{col} \leftarrow \{i\frac{N}{a} + j + \ell d : 0 \leq \ell < c\}$
5:    $\mathbf{Y}[:, \texttt{row}] \leftarrow \mathbf{Y}[:, \texttt{row}] + \mathbf{X}[:, \texttt{col}]\,\mathbf{K}^\top[\texttt{col}, \texttt{row}]$
6: **end for**

---

**Why Algorithm 2 is Mathematically Equivalent to Algorithm 1?** Given a KS pattern $\boldsymbol{\pi} = (a, b, c, d)$, the corresponding support $\mathbf{S}_{\boldsymbol{\pi}} := \mathbf{I}_a \otimes \mathbf{1}_{b \times c} \otimes \mathbf{I}_d$ has shape $abd \times acd$. We partition its row indices as $[\![0, abd - 1]\!] := \{0, \ldots, abd - 1\} = \bigcup_{i=0}^{a-1} \bigcup_{j=0}^{d-1} \texttt{row}_{i,j}$ where:

$$\texttt{row}_{i,j} := \left\{ i\frac{M}{a} + j + kd : 0 \leq k < b \right\}. \quad (1)$$

Due to the structure of $\mathbf{S}_{\boldsymbol{\pi}}$, all $b$ rows in each $\texttt{row}_{i,j}$ share the same support, which we denote $\texttt{col}_{i,j}$:

$$\texttt{col}_{i,j} := \left\{ i\frac{N}{a} + j + \ell d : 0 \leq \ell < c \right\}. \quad (2)$$

The family $\{\texttt{col}_{i,j}\}_{(i,j)}$ forms a partition of the column indices $[\![0, acd - 1]\!]$. This means the matrix multiplication $\mathbf{Y} = \mathbf{X}\mathbf{K}^\top$, when restricted to the rows in $\texttt{row}_{i,j}$, can be expressed as:

$$\begin{aligned} \mathbf{Y}[:, \texttt{row}_{i,j}] &= \mathbf{X}\left(\mathbf{K}^\top[:, \texttt{row}_{i,j}]\right) \\ &= \mathbf{X}[:, \texttt{col}_{i,j}]\,\mathbf{K}^\top[\texttt{col}_{i,j}, \texttt{row}_{i,j}], \end{aligned} \quad (3)$$

where $\mathbf{M}[:, J]$ denotes matrix $\mathbf{M}$ restricted to columns $J$, and $\mathbf{M}[I, J]$ denotes the submatrix of $\mathbf{M}$ with rows $I$ and columns $J$. By grouping rows that share the same support, the full KS matrix multiplication reduces to a set of smaller, independent dense multiplications—one per $\texttt{row}_{i,j}$—that can be processed in parallel. This is illustrated in Figure 5. Hence, Algorithm 2 is mathematically equivalent to Algorithm 1.

### 4.2. New Kernel Implementation

We now introduce our fused implementation of Algorithm 2, denoted KERNEL in the remainder. A compact pseudocode sketch is given in Algorithm 3.

**KERNEL versus the permutation-based baseline.** While both Algorithms 1 and 2 ultimately compute the same set of dense sub-products—one per tile $\texttt{row}_{i,j}$—they differ in how they reach them, and hence in their natural GPU implementations:

• **Algorithm 1.** This baseline performs two explicit permutations of $\mathbf{X}$ and $\mathbf{Y}$ to map the entries of $\mathbf{X}[:, \texttt{col}_{i,j}]$ and $\mathbf{Y}[:, \texttt{row}_{i,j}]$—whose columns are spaced by $d$— to contiguous regions in the permuted tensors $\tilde{\mathbf{X}}$ and $\tilde{\mathbf{Y}}$. This enables the use of standard dense GEMM kernels, which require contiguity in global memory. In practice, this strategy is implemented in BMM via two additional PERMUTE kernels, which rewrite each tile contiguously in *global memory*.

• **Algorithm 2.** This variant avoids global permutations altogether. It computes each dense sub-product directly, even when the corresponding rows and columns in $\texttt{row}_{i,j}$ and $\texttt{col}_{i,j}$ are not contiguous in global memory. Our fused implementation KERNEL follows this approach: each thread block directly loads $\mathbf{X}[:, \texttt{col}_{i,j}]$ and $\mathbf{K}^\top[\texttt{col}_{i,j}, \texttt{row}_{i,j}]$ from global to shared memory, performs the multiplication, and writes the result into $\mathbf{Y}[:, \texttt{row}_{i,j}]$, thus bypassing the two extra permutation passes.

**Thread-block tiling.** We assign to each thread block a single dense sub-product $\mathbf{X}[:,\texttt{col}_{i,j}] \cdot \mathbf{K}^\top[\texttt{col}_{i,j},\texttt{row}_{i,j}]$. We make threads within a block:

(i) cooperatively load the two tiles from global memory to shared memory,

(ii) move their assigned subtiles to registers,

(iii) perform multiply-and-accumulate operations,

(iv) store intermediate results to shared memory, and

(v) write cooperatively the final output tile back to global memory.

Standard CUDA optimizations such as double buffering, vectorized memory access, and warp-level parallelism are applied (see Appendix D.3 for implementation details).

**Read/write efficiency.** Replacing two permutation kernels with direct memory access leads to different tradeoffs depending on the layout:

- **KS matrix tiles.** As in BMM, the factor $\mathbf{K}$ is preprocessed and stored so that every tile $\mathbf{K}^\top[\texttt{col}_{i,j},\texttt{row}_{i,j}]$ is already contiguous in global memory. This setup cost is paid once and amortized across inference runs with different inputs $\mathbf{X}$, so we exclude it from our analysis (Section 1).

- **Input/output tiles.** Contiguity of memory access now depends on the tensor layout. In the feature-major layout (*batch-size-last* in row-major), each column of $\mathbf{X}[:,\texttt{col}_{i,j}]$ and $\mathbf{Y}[:,\texttt{row}_{i,j}]$ is stored contiguously in global memory, enabling efficient coalesced access within columns. In this setting, KERNEL benefits from *both eliminating permutations and maintaining memory locality*.

  In contrast, under the batch-major layout (*batch-size-first* in row-major), entries within the same column of $\mathbf{X}[:,\texttt{col}_{i,j}]$ or $\mathbf{Y}[:,\texttt{row}_{i,j}]$ are not contiguous in global memory, as the indices in $\texttt{col}_{i,j}$ and $\texttt{row}_{i,j}$ are spaced by $d$. This leads to strided, non-coalesced access patterns. While KERNEL still avoids the overhead of explicit permutations, this benefit has to be weighted against more fragmented memory access.

In summary, KERNEL eliminates two global-memory round trips compared to permutation-based implementations. It is often faster in the *batch-size-last* layout, where it benefits from both avoiding explicit permutations and preserving coalesced memory access. In the *batch-size-first* layout, this advantage must be weighed against the cost of non-coalesced memory access due to strided reads and writes; nevertheless, KERNEL remains competitive, and even outperforms baselines in several cases, as observed in Section 5.

**Algorithm 3** Sketch of the fused **output-stationary** kernel (one tile ($\texttt{row}_{i,j}$, $\texttt{col}_{i,j}$) assigned to each thread block).

**Require:** pattern $\boldsymbol{\pi} = (a,b,c,d)$, inputs $\mathbf{X} \in \mathbb{R}^{B \times acd}$, KS factor $\mathbf{K} \in \mathcal{K}_{\boldsymbol{\pi}}$
**Ensure:** output $\mathbf{Y} := \mathbf{X}\mathbf{K}^\top \in \mathbb{R}^{B \times abd}$
1: identify $(i,j)$ tile assigned to current thread block
2: compute $\texttt{row}, \texttt{col}$ as in Algorithm 2
3: **global $\to$ shared**: load $\mathbf{X}[:,\texttt{col}], \mathbf{K}^\top[\texttt{col},\texttt{row}]$
4: **for** stride $=0$ to $c-1$ by size-subtile **do**
5:     identify subtile assigned to current thread
6:     **shared$\to$registers**: load subtiles of $\mathbf{X}[:,\texttt{col}]$ and $\mathbf{K}[\texttt{row},\texttt{col}]$
7:     Multiply and accumulate into registers
8:     Prefetch next subtile in parallel (double buffering)
9: **registers$\to$global**: store $\mathbf{Y}[:,\texttt{row}]$

### 4.3. How Much Traffic Does the Fuse Save?

**Against Specialized Baselines.** Recall $\#\text{nz} = abcd$ is the number of nonzeros of a KS matrix with pattern $(a,b,c,d)$, $B$ is the batch size, and $N = acd$, $M = abd$ are the input and output dimensions. Specialised baselines call three kernels (PERMUTE-GEMM-PERMUTE). The first PERMUTE involves $2BN$ global memory operations (read $\mathbf{X} \in \mathbb{R}^{B \times N}$ once, write its permuted version $\tilde{\mathbf{X}}$ once). The GEMM involves $BN + BM$ global memory operations (read once the permuted inputs $\tilde{X}$, write once the permuted outputs $\tilde{Y}$). The last PERMUTE involves $2BM$ global memory operations of the input and output tensors (read $\tilde{\mathbf{Y}}$ once, write $\mathbf{Y}$ once). The total number of global reads and writes on the input and output tensors is $3B(N + M)$. Our kernel moves each entry exactly once, saving the extra $2B(N+M)$ global reads/writes of the two permutations. The ratio of wasted global memory traffic to useful multiplies is therefore twice

$$\frac{B(N+M)}{B\,\#\text{nz}} \;=\; \frac{b+c}{bc} \;=\; h(b,c), \qquad (4)$$

which yields the heuristic that correlated with the time spent on permutations in Figure 4. Since our new kernel precisely reduces the cost of the permutations, we expect better gains for patterns with large $h(b,c)$, which is indeed what we observe in Section 5.

**Against Generic Baselines.** Generic dense/CSR PyTorch baselines, which ignore the KS structure, are noted DENSE and SPARSE. Compared with DENSE, we skip all zero entries of $\mathbf{K}$, avoiding unnecessary computations. Compared with SPARSE, we match the number of global reads and writes but enjoy better coalescing thanks to knowing the KS support.

**Take-Away:** *The fused, output-stationary* KERNEL *reduces global reads and writes of the input and output tensors by up to a factor three, directly translating into the speedups and energy cuts in practice, as we now report the next section.*

*Table 2.* Across the 600 patterns, how often is an algorithm faster and by how much (median speedup in parentheses), in FP32.

| Comparison | Win rate | Median $\times$ faster |
|---|---|---|
| $\min\{$KERNEL, BMM, EINSUM, BSR$\}$ $< \min\{$DENSE, SPARSE$\}$ | 98.1 % | $\times 6.5$ |
| BMM $< \min\{$EINSUM, BSR, DENSE, SPARSE$\}$ | 90.0 % | $\times 1.36$ |
| KERNEL $< \min\{$BMM, EINSUM, BSR, DENSE, SPARSE$\}$ | 85.5 % | $\times 1.39$ |

# 5. Benchmark: 600 KS patterns, Time & Energy

We now quantify the benefit of KERNEL over the fastest public baselines—BMM, BSR, and EINSUM—on a single Nvidia A100 (40 GB) for time and on a single Nvidia V100 (32 GB) for energy[4].

**Benchmarked KS Patterns.** We explore patterns of the form $\boldsymbol{\pi} = (a, b, c, d)$, sweeping over the space $\alpha \times \beta \times \beta \times \alpha$, with $\alpha = \{1, 2, 3, 4, 6, 8, 12, 16, 24, 32, 48, 64, 96, 128\}$ and $\beta = \{48, 64, 96, 128, 192, 256, 384, 512, 768, 1024\}$. We constrain the shapes by enforcing $b = c$, $b = 4c$, or $c = 4b$, which reflect common configurations in Transformer architectures.[5] This results in over **600** distinct KS matrix configurations, with sizes ranging from $102 \times 102$ up to $131072 \times 131072$—spanning six orders of magnitude and broadly covering[6] the hidden dimensions found in vision and language Transformers. The input batch size is fixed to $B = 128 \times 196 = 25088$, corresponding to a batch of 128 sequences with a context length of 196 tokens, which is a typical value encountered in Transformer inference.[7] All results below take, for every implementation, the most efficient (in time or energy) of *batch-size-first* and *batch-size-last* layouts; the full split is in Appendix E.6.

**Covered Sparsity Levels.** The 600 KS patterns selected for this benchmark are skewed toward high sparsity: the *median sparsity* is 97.9 %, and 75 % of the patterns have sparsity $\geq 91.7$ %. Additional details on the sparsity distribution are provided in Appendix E.2.

---

[4]`pyJoules` is not yet compatible with Nvidia A100 GPUs so we benchmarked the energy consumption on V100 GPUs. Note that we observed the same *relative* time efficiency ranking on the two GPUs.

[5]Recall that for a KS pattern $(a, b, c, d)$, the input and output dimensions of the matrix are given by $N = acd$ and $M = abd$. When $b = c$, we have $N = M$, yielding a square matrix suitable for weight matrices in the self-attention block. When $b = 4c$ (resp. $c = 4b$), the output dimension becomes $M = 4N$ (resp. $M = N/4$), matching the shape of up-projection (resp. down-projection) layers typically found in feed-forward blocks.

[6]The largest LLaMA 3 model (405B) includes FFN layers with matrices up to $53\,248 \times 53\,248$.

[7]e.g., in ViT with a patch size of $16 \times 16$ for image resolution of $224 \times 224$

## 5.1. Time: KERNEL Wins on 85 % of Patterns in FP32

Table 2 highlights three key findings: (i) KS-aware implementations (KERNEL, BMM, EINSUM, BSR) are more than $\times 6$ faster than generic baselines that ignore the KS structure (DENSE, SPARSE); (ii) among existing methods, BMM is the fastest baseline (excluding our new KERNEL from the comparison); (iii) our fused, output-stationary KERNEL surpasses all baselines on 85 % of the benchmarked grid in FP32, achieving a median speedup of $\times 1.4$.

Because the benchmark grid includes many highly sparse KS patterns, we also compare performance at fixed sparsity levels in Appendix E.2. This analysis confirms that KERNEL provides consistent speedups across *all sparsity regimes*—not just in the extremely sparse or dense cases.

## 5.2. What Drives the Speedup? Mostly the Heuristic $h(b, c) = \frac{b+c}{bc}$

To understand what drives the speedup achieved by KERNEL, we perform a multiple linear regression[8] in log-log scale using two predictors: the density[9] $\frac{1}{ad}$ and the heuristic $h(b, c) = \frac{b+c}{bc}$. Before fitting the regression, we verified that the two predictors—density and the heuristic $h(b, c)$—are only weakly correlated across the 600 KS patterns (Pearson correlation: –0.28). This low correlation ensures that the regression is not affected by multicollinearity, and both predictors can be interpreted as contributing distinct information to the model.

Letting speedup := $\frac{\min \text{time}(\text{BMM},\text{BSR},\text{EINSUM})}{\text{time}(\text{KERNEL})}$, the regression yields:

$$\log(\text{speedup}) \approx 1.69 - 0.031 \log(\text{density}) + 0.325 \log(h).$$

This model achieves an adjusted $R^2$ of 0.697 and passes the Jarque–Bera normality test[10] on residuals ($p = 0.12$). Both predictors are statistically significant ($p < 0.001$).

Notably, $h(b, c)$ is approximately $\times 10$ more influential than density in explaining the speedup. To further isolate the impact of the heuristic $h(b, c)$ from sparsity, we fix the sparsity and examine how the speedup varies with $h(b, c)$.

---

[8]We excluded fully dense patterns from this regression, i.e., those with density = 1.

[9]We chose density over sparsity here, as it gave better regression results.

[10]We were able to pass the normality test in log-log scale, but not in linear-linear or log-linear settings.

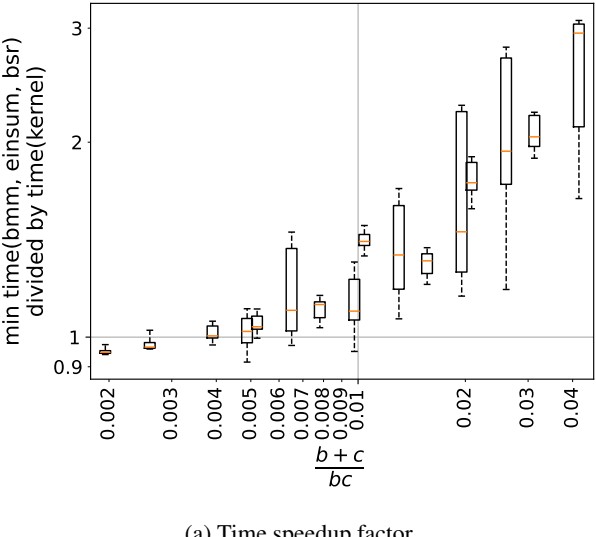

(a) Time speedup factor.

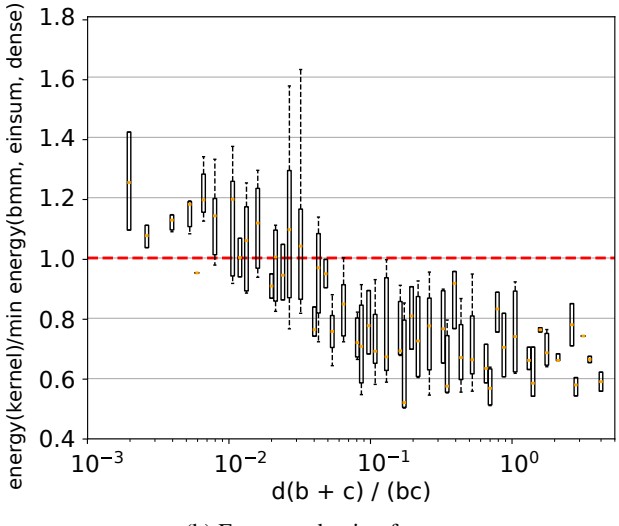

(b) Energy reduction factor.

*Figure 6.* Comparison of kernel in terms of (a) times speed up factor and (b) energy consumption, relative to $\min(\text{BMM}, \text{EINSUM}, \text{BSR})$, on all the 600 KS patterns in FP32. For each implementation, we take the minimum time/energy spent between the *batch-size-first* and *batch-size-last* memory layouts. We regroup patterns by their value of $h(b,c) = \frac{b+c}{bc}$ for(a) and $d \times h(b,c)$ for (b).

For each sparsity bin in Appendix E.2, we observe a strong and consistent correlation in log-log scale (between 0.76 and 0.91), confirming that $h(b,c)$ is a reliable predictor of performance regardless of sparsity; see Appendix E.2 for details.

**In conclusion,** *maximizing $h(b,c)$ is a simple and effective design principle for selecting KS patterns that benefit from* KERNEL's speedup over prior baselines (see Figure 6a). In contrast, density has only a weak and inconsistent influence once $h$ is controlled for.

### 5.3. Energy: 15 % Median Saving in FP32, Predicted by $d \times h(b,c)$

Because each column gathered by a permutation is spaced by $d$, the *distance* travelled by memory requests grows with $d$, and $d \times h(b,c)$ is a good proxy for the proportion of energy spent on memory operations. Figure 6b matches theory: the higher that proxy, the more expensive the rewrites and the larger the energy saving of KERNEL. Overall, KERNEL reduces energy consumption compared to existing baselines on 72 % of the patterns, achieving a median saving of 15 % in FP32.

### 5.4. Additional Remarks

**BSF vs. BSL.** Switching baselines to *batch-size-last* (BSL) does not change the runtime of BMM[11] However, KER-NEL gains a stable $\times 2$ when switching from *batch-size-first*

---

[11]Note that BMM remains the best baselines compared to EIN-SUM, BSR both in BSF and BSL layouts (Appendix E.6 for details).

(BSF) to *batch-size-last*, thanks to fully coalesced writes as we saw in Section 4.2; full numbers are in Appendix E.6.

**Impact of Size.** Speedups increase with matrix size $M \times N = abd \times acd$ (Figure 12); our largest test case ($131k \times 131k$) already exceeds Llama 3-405B's hidden layers.

**Portability.** All results above use CUDA. For broader hardware support, we also provide an OpenCL version, enabling benchmarks on non-NVIDIA devices like AMD GPUs and CPUs.

### 5.5. Take-Away

*Exploiting the KS structure is essential: $\times 6$ vs. dense/CSR generic baselines. Removing the two permutations gives in float-precision an additional $\times 1.4$ in time and $15\%$ energy reduction. The simple rule "maximise $h(b,c)$" predicts where the fused kernel shines; sparsity alone does not.*

## 6. Broader Impact: End-to-End Gains on ViT and GPT Models

Industry reports estimate that inference accounts for over 90 % of the total cost of machine learning at scale (HPCwire, 2019; Barr, 2019). To test whether the layer-wise speedups of KERNEL from Section 5 translate into application-level gains, we inject KS sparsity into a Vision Transformer (Dosovitskiy et al., 2020) and a GPT-style language model (Radford et al., 2019), and measure inference time.

**Target Architecture.** Fully connected (FC) layers are natural candidates for KS-based acceleration. In our

experiments, we simply replace the standard PyTorch `Linear` module with our publicly available `KSLinear` module.[12] Transformer architectures such as ViT and GPT offer a suitable testbed, as they contain numerous FC layers—accounting for 30–60% of total inference time depending on model size (Appendix E.7).[13] We evaluate our method on ViT-S/16 (21M parameters) and GPT-2 Medium (345M), which both fit comfortably on a single GPU.

**Setup.** Following the protocol of Dao et al. (2022b), we replace specific weight matrices of linear layers in ViT and GPT with a *product of two* KS factors, using patterns detailed in the appendices. All other components of the Transformer forward pass use the default PyTorch kernels in the default *batch-size-first* layout. Complete setup details are provided in Appendix E.8.

**Results.** Table 3 shows that the fused kernel reduces end-to-end latency by **22 %** on ViT-S/16 in FP32, corresponding to a ×1.28 speedup over the original dense implementation. For comparison, the best public baseline achieves an 11 % reduction (×1.12 speedup). On GPT-2 Medium, the fused kernel yields a **16 %** latency reduction (×1.19 speedup), while the best baseline achieves 12 % (×1.14 speedup). Additional experimental results are provided in Appendix E.8.

*Table 3.* Latency ratio versus the dense baseline (`fully-connected`). Lower is better.

|  | Sparsity level | $\frac{\text{time}(\text{BMM})}{\text{time}(\text{DENSE})}$ | $\frac{\text{time}(\text{KERNEL})}{\text{time}(\text{DENSE})}$ |
|---|---|---|---|
| ViT-S/16 | 57.1 % | 0.89 | **0.78** |
| GPT-2 Medium | 24.8 % | 0.88 | **0.84** |

**Discussion.** In our experiments, inference in ViT and GPT models is run using the *batch-size-first* layout. However, as discussed in Section 5.4, switching to *batch-size-last* can yield up to a ×2 speedup within KS layers. Extending *batch-size-last* support to other core components of neural network inference—such as activation functions, softmax, and LayerNorm—could unlock additional gains, provided these operations can be implemented with similar efficiency in *batch-size-last*. Assessing the end-to-end impact of *batch-size-last* on full-model performance remains an open engineering challenge.

**Take-Away:** *Replacing dense weights by KS factors and using the new KERNEL speedups the end-to-end inference of Transformers in FP32, suggesting that the proposed tiling strategy can bring practical benefits in real-world settings.*

---

[12] Available at github.com/PascalCarrivain/ksmm.

[13] See Appendix E.8 for why applying KS sparsity to convolutional networks poses additional challenges.

# 7. Conclusion and Discussion

We revisited GPU multiplication with Kronecker-sparse (KS) matrices, identified that permutation overheads dominate existing kernels, and proposed an output-stationary tiling strategy that fuses both permutations with the GEMM into a single CUDA/OpenCL kernel. Across 600 KS patterns, our kernel is ×1.4 faster (median, FP32) than the best public baseline and reduces energy consumption by 15 %. When integrated into ViT and GPT models, it yields practical inference-time speedups in FP32.

## 7.1. Key Take-Aways

**Fused Tiling Strategy.** Our output-stationary tiling reduces memory traffic by fusing both permutations with the GEMM in a single kernel.

**Simple Design Rule.** A one-line heuristic $h(b, c) = \frac{b+c}{bc}$ helps identify patterns well-suited for this kernel.

**We release a `KSLinear` PyTorch module** supporting backend selection and both *batch-size-first* and *batch-size-last* memory layouts, and which can be used in place of standard PyTorch linear layers.

## 7.2. Limitations and Next Steps

(i) Gains obtained in FP32 are not as large in FP16/FP8 where tensor cores become available (see Appendix E.9 for the analysis of FP16). This calls for further investigations: is it our kernel that can be further improved, or the other implementations that have been particularly optimized in these regimes?

(ii) *Batch-size-last* layout outperforms *batch-size-first* in our KERNEL (Figure 14), but PyTorch defaults to BSF. Using BSL in BSF pipelines requires costly transpositions that may offset the gains. Our `KSLinear` layer supports both: BSF for drop-in use, and BSL for higher performance when compatible. More broadly, the benefits of BSL raise an open question: how does batch dimension placement affect neural network efficiency in general?

(iii) An interesting direction is to extend end-to-end latency experiments to larger models than those in Section 6, and to study the trade-off between inference gains and accuracy when training/finetuning with `KSLinear` layers.

(iv) Cache behavior was not analyzed in depth; future work may reveal strengths or limitations of our tiling strategy and suggest further improvements.

(v) Speedups depend on batch size, KS pattern, and a few kernel hyperparameters (e.g., thread count, tile shape). Tuning is setup-specific and tedious, so we provide presets for the 600 patterns we benchmarked and leave general tuning strategies to future work.

(vi) Beyond NVIDIA hardware, the OpenCL port opens the door to studies on AMD GPUs and other platforms.

## Acknowledgments

This work was supported in part by the AllegroAssai ANR-19-CHIA-0009, by the NuSCAP ANR-20-CE48-0014 projects of the French Agence Nationale de la Recherche and by the SHARP ANR project ANR-23-PEIA-0008 in the context of the France 2030 program.

Tung Le was supported by AI Interdisciplinary Institute ANITI funding, through the French "Investments for the Future – PIA3" program under the grant agreement ANR-19-PI3A-0004, and Air Force Office of Scientific Research, Air Force Material Command, USAF, under grant numbers FA8655-22-1-7012.

The authors thank the Blaise Pascal Center for the computational means. It uses the SIDUS (Quemener & Corvellec, 2013) solution developed by Emmanuel Quemener. We would also like to thank Patrick Pérez, Gilles Puy, Elisa Riccietti, Nicolas Brisebarre, and Rémi Gribonval for their useful feedback, and Emmanuel Quemener for reserving computing resources for us while we ran our experiments.

## Impact Statement

This paper presents work whose goal is to advance the field of Machine Learning. There are many potential societal consequences of our work, none which we feel must be specifically highlighted here.

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

# Appendices

**Lay summary.** Today's AI models are costly to run because they spend most of their time and energy multiplying huge tables of numbers. One way to speed this up is to use tables with lots of zeros, since multiplying by zero does nothing and can be skipped. If we know where the zeros are in advance, we can organize the work to skip them efficiently—but how we skip them matters.

In this work, we focus on a promising zero pattern that has been widely studied. We show that, even with this pattern, the fastest current Graphical Processing Unit (GPU) methods waste up to half their time just shuffling numbers around before doing the real calculations. While this shuffling is meant to help (to reorganize the table in big chunks that can be skipped), our work reveals that it is unnecessary, and we propose a faster workaround.

By keeping the table data in place and changing how the GPU handles the work, we skip the zeros without extra shuffling. Our method runs up to $40\,\%$ faster and uses $15\,\%$ less energy across 600 test cases. We provide open-source code for both GPUs and CPUs, plus a one-line rule of thumb to tell when it helps. Plugged into large AI models like Transformers, it cuts total running time by up to 22%, paving the way for faster, cheaper AI.

## A. Existing GPU Implementations of Kronecker-Sparse Matrix Multiplication

This section describes the existing GPU implementations for KS matrix multiplication on PyTorch. All source code is available in the forward pass of the `KSLinear` module we release at [github.com/PascalCarrivain/ksmm](github.com/PascalCarrivain/ksmm).

### A.1. Specialized Baselines

**BMM (batched GEMM).** The implementation released with the "Monarch" layers of Dao et al. (2022b)[14] permutes the weight matrix once offline and stores $\tilde{\mathbf{K}} \in \mathbb{R}^{ad \times b \times c}$ as a dense `float`/`half` tensor of shape $(ad, b, c)$. Given a batch of inputs $\mathbf{X} \in \mathbb{R}^{B \times acd}$, the forward pass executes Algorithm 1 with fast tensor reshapes for the input and output permutations, and with the fast routine `torch.bmm` for the multiplication with $\tilde{\mathbf{K}}$:

1) reshape $\mathbf{X}$ so that multiplying by $\mathbf{Q}^\top$ is a view;
2) perform the batched GEMM `torch.bmm`;
3) reshape the result so that multiplying by $\mathbf{P}^\top$ is a view.

A concise PyTorch version follows (layout *batch-size-first*; swap first/last dimensions for *batch-size-last*), where we denote by `K_bmm` the 4D PyTorch tensor storing $\tilde{\mathbf{K}}$ and by `T_bsf` the *batch-size-first* PyTorch tensor corresponding to a matrix $\mathbf{T}$:

```
def kronecker_bmm(X_bsf, K_bmm):
    batch_size = X_bsf.shape[0]
    X_perm = (
        X_bsf.view(batch_size, a, c, d)
        .transpose(-1, -2)
        .reshape(batch_size, a * d, c)
        .contiguous()
        .transpose(0, 1)
    )
    Y_perm = torch.empty(batch_size, a * d, b).transpose(0, 1)
    Y_perm = torch.bmm(X_perm, K_bmm.transpose(-1, -2))
    Y_bsf = (
        Y_perm.transpose(0, 1)
        .reshape(batch_size, a, d, b)
        .transpose(-1, -2)
        .reshape(batch_size, a * b * d)
    )
    return
```

**BSR (block-sparse GEMM).** The same algebraic steps apply, but $\tilde{\mathbf{K}}$ is stored once for all in **B**lock-compressed **S**parse **R**ow format as a tensor `K_bsr`, and Step (2) calls `torch.sparse.mm`:

```
def kronecker_bsr(X_bsf, K_bsr):
```

---

[14]The original release supported only $\boldsymbol{\pi} = (a, b, c, d)$ with $a=1$ or $d=1$; we extended it to the general case.

```
2      batch_size = X_bsf.shape[0]
3      X_perm = (
4          X_bsf.view(batch_size, a, c, d)
5          .transpose(-1, -2)
6          .reshape(batch_size, a * c * d)
7      )
8      Y_perm = torch.nn.functional.linear(
9          X_perm, K_bsr
10     )
11     Y_bsf = (
12         Y_perm.view(batch_size, a, d, b)
13         .transpose(-1, -2)
14         .reshape(batch_size, a * b * d)
15     )
16     return Y_bsf
```

EINSUM **(tensor contraction).** Following the public prototype in the Monarch repository (Dao et al., 2022b), the $abcd$ nonzeros of $\mathbf{K}$ are packed in $K_{\text{einsum}} \in \mathbb{R}^{a \times b \times c \times d}$. After reshaping the input $\mathbf{X} \in \mathbb{R}^{B \times acd}$ to $X_{\text{einsum}} \in \mathbb{R}^{B \times a \times c \times d}$ and the output $\mathbf{Y} \in \mathbb{R}^{B \times abd}$ to $Y_{\text{einsum}} \in \mathbb{R}^{B \times a \times b \times d}$, the contraction $(Y_{\text{einsum}})_{:,a,b,d} = \sum_c (X_{\text{einsum}})_{:,a,c,d} (K_{\text{einsum}})_{a,b,c,d}$ is issued via einops.einsum:

```
1  def kronecker_einsum(X_bsf, K_einsum):
2      X_perm = einops.rearrange(X_bsf, "... (a c d) -> ... a c d", a=a, c=c, d=d)
3      Y_perm = einops.einsum(X_perm, K_einsum, "... a c d, a b c d -> ... a b d")
4      Y_bsf = einops.rearrange(Y_perm, "... a b d-> ... (a b d)")
5      return Y_bsf
```

The second line of this code does at the same time all the matrix multiplications $\mathbf{Y}[:, \text{row}] \leftarrow \mathbf{X}[:, \text{col}]\mathbf{K}^\top[\text{col}, \text{row}]$ for all the pairs $(\text{row}, \text{col})$ in Algorithm 2.

**Other KS implementations** The "Monarch" batched-GEMM code already covered above (BMM) is the maintained reference for Butterfly/Monarch layers of Dao et al. (2019; 2022b). We could not compile the original Butterfly Transform kernels (Dao et al., 2019; Vahid et al., 2020) on recent CUDA; an in-house re-implementation turned out much slower than other baselines and is therefore omitted from the benchmark.

## A.2. Generic Baselines

DENSE. All $MN$ entries of $\mathbf{K}$ are stored (even zero entries) and the forward pass calls torch.nn.functional.linear. With *batch-size-first* layout this is the PyTorch default; in *batch-size-last* we select torch.matmul after an internal benchmark of alternatives.

SPARSE. Only the $abcd$ non-zeros are kept (CSR format) and the same linear/matmul entry point is used. This baseline uses the sparsity but not its specific *Kronecker* structure.

## A.3. Prior Empirical Reports on the Efficiency of Existing KS Matrix Multiplication Algorithms

Dao et al. (2022b) reported a $\times 2$ training speedup on image classification and language-modeling tasks by replacing dense layers with products of two KS factors.

Fu et al. (2023) measured a speedup exceeding $\times 2$ at inference for FFT-based layers $\mathbf{X} \mapsto \mathbf{W}^{-1}(\mathbf{K} \odot \mathbf{W}\mathbf{X})$, where $\mathbf{K}$ is dense, $\odot$ is element-wise multiplication, and $\mathbf{W}$ is the DFT matrix in dimension $\geq 4096$. These gains are relevant to KS matrices since the fast implementation of the DFT (the FFT) is equivalent to a sequential multiplication with Kronecker-sparse matrices (see Figure 1).

Our study complements the above by isolating the most elementary operation in these different pipelines: the multiplication by a *single* KS factor $\mathbf{X} \mapsto \mathbf{K}\mathbf{X}$. We benchmark more than 600 distinct KS patterns spanning six orders of magnitude in matrix shapes, comparing existing GPU baselines on PyTorch.

# B. Connections to Other Sparsity Schemes

## B.1. Kronecker-Sparsity in the Wider Sparsity Landscape

*Which sparsity patterns studied in prior work are instances of Kronecker-structured designs?*

Table 4 shows that several sparsity patterns previously proposed for neural network compression can be expressed as special cases of KS patterns, illustrating the broad applicability of the framework studied in this paper.

*Table 4.* KS patterns used in prior work to replace dense layers in neural networks. For a product $\mathbf{K}_1 \ldots \mathbf{K}_L$ we list $(a_\ell, b_\ell, c_\ell, d_\ell)_{1 \leq \ell \leq L}$.

| | MATRIX SIZE | KRONECKER PATTERNS |
|---|---|---|
| DENSE | $M \times N$ | $(1, M, N, 1)$ |
| LOW-RANK | $M \times N$ | $(1, M, r, 1), (1, r, N, 1)$ |
| SQUARE DYADIC (DAO ET AL., 2019; VAHID ET AL., 2020) | $2^L \times 2^L$ | $(2^{\ell-1}, 2, 2, 2^{L-\ell})_{\ell=1}^L$ |
| KALEIDOSCOPE (DAO ET AL., 2022B) | $2^L \times 2^L$ | $(2^{\ell-1}, 2, 2, 2^{L-\ell})_{\ell=1}^L \cup (2^{L-\ell}, 2, 2, 2^{\ell-1})_{\ell=1}^L$ |
| BLOCK BUTTERFLY (DAO ET AL., 2022A) | $2^L t \times 2^L t$ | $(2^{\ell-1}, 2t, 2t, 2^{L-\ell})_{\ell=1}^L$ |
| MONARCH (DAO ET AL., 2022B; FU ET AL., 2023) | $M \times N$ | $(1, M/p, \min(M, N)/p, p), (p, \min(M, N)/p, N/p, 1)$ |
| DEFORMABLE BUTTERFLY (LIN ET AL., 2021) | $a_1 b_1 d_1 \times a_L c_L d_L$ | $(a_\ell, b_\ell, c_\ell, d_\ell)_{\ell=1}^L$ WITH $a_\ell c_\ell d_\ell = a_{\ell+1} b_{\ell+1} d_{\ell+1}$. |

*How does the Kronecker-sparsity pattern compare to other forms (unstructured or structured) of sparsity and when is each preferable?*

Structured patterns–Kronecker, block, channel-wise, ...–tend to win on latency and energy as the prior knowledge of the sparsity pattern enables the design of more efficient algorithms for inference on GPU. Beyond implementation, structured sparsity gives stronger theoretical guarantees. For example, the associated function space is closed and a minimizer of the loss is guaranteed to exist, unlike in the unstructured case where solutions may diverge toward non-attaining infima (Theorem 4.2 in (Le et al., 2023)). That said, more specific performance comparisons ultimately depend on the particular task, hardware, model, and sparsity structure considered.

## B.2. Dense→KS Conversion (Out of Scope in this Paper)

*How can one convert a dense layer to a Kronecker-sparse one and what is its computational cost?*

Given a dense matrix $\mathbf{W}$ (e.g., a weight matrix obtained from training), one can learn factors $\mathbf{K}_1 \cdots \mathbf{K}_L$ such that $\mathbf{W} \approx \mathbf{K}_1 \cdots \mathbf{K}_L$ (in Frobenius norm) using a quasi-optimal butterfly factorization algorithm (with provable error guarantees) that has a roughly $\mathcal{O}(MN)$ time complexity for the factorization of a matrix of size $M \times N$ (Le et al., 2024; Zheng et al., 2023; Le et al., 2022). This factorization algorithm is implemented in the `lazylinop` package (see Appendix D.1 for details about this package). In particular, it is shown that this factorization algorithm is more accurate and efficient than alternating least squares (Lin et al., 2021) or gradient descent (Dao et al., 2019). In our work, we focus on pure inference mode where the KS matrix $\mathbf{K}$ is given and fixed. The cost of a potential upstream conversion is therefore ignored in our study.

## B.3. Analogy with Sparse 3D Convolutions

An interesting analogy can be drawn between the positioning of our new implementation relative to existing GPU-based KS matrix multiplication algorithms, and how recent works on sparse 3D convolutions (Tang et al., 2023; Spconv Contributors, 2022; Yan et al., 2018) relate to TorchSparse (Tang et al., 2022).

TorchSparse (Tang et al., 2022) performs sparse 3D convolutions in three stages: GATHER-MATMUL-SCATTER. The works (Tang et al., 2023; Spconv Contributors, 2022; Yan et al., 2018) improved performance by overlapping memory transfers with computation. Similarly, for the sparse problem we tackle, our new dataflow strategy implementing Algorithm 2 leads to a comparable improvement: input/output permutations and matrix multiplication can now be fused into a *single* kernel (Figure 3, right), enhancing efficiency.

That said, the analogy seems to stop there. The underlying structures of the two problems are quite different, and it does not seem that there is a straightforward way to adapt the tiling strategy from Tang et al. (2023) to the nested Kronecker layout we study.

### B.4. Relation to Sparse-Tensor Compilers

State-of-the-art sparse-tensor compilers, e.g., SparseTIR (Ye et al., 2023) or Fractal (Guan et al., 2024), are primarily designed for conventional sparse formats such as CSR or BSR, combined with regular block tilings. KS matrices, by contrast, exhibit a nested sparsity pattern: a block-diagonal layout with $b \times c$ sub-blocks, each refined internally by a $d \times d$ identity structure. Representing such a hierarchy would require support for an outer block-sparse format with an inner identity pattern—capabilities that current compilers do not offer natively.

Furthermore, while our Algorithm 2 uses standard tiling strategies when possible, it sometimes favors non-contiguous tile shapes along certain axes to reduce memory transfers and improve efficiency. This form of layout flexibility is not yet supported by existing sparse compilers, to the best of our knowledge.

Until compiler infrastructure evolves to support nested sparsity and more flexible tiling schemes, hand-optimized kernels seems to remain an effective solution for running KS workloads.

## C. Perfect-Shuffle Permutations Used in Algorithm 1

In practice, existing implementations of Algorithm 1 select the permutation matrices

$$\mathbf{P} := \big(\mathbf{I}_a \otimes \mathbf{P}_{b,d}\big), \quad \mathbf{Q} := \mathbf{I}_a \otimes \mathbf{P}_{c,d}, \tag{5}$$

where $\mathbf{P}_{p,q}$ is the $(p, q)$ *perfect-shuffle* permutation matrix. This appendix reviews perfect-shuffle permutations and shows why the choice in (5) produces the permuted matrix

$$\tilde{\mathbf{K}} := \mathbf{P}^\top \mathbf{K} \mathbf{Q}^\top$$

that is *block-diagonal with dense sub-blocks*—specifically, $ad$ diagonal dense blocks of size $b \times c$. Because each block can be multiplied with highly optimized dense routines, Step (2) of Algorithm 1 becomes fast, motivating the use of these particular permutations.

### C.1. Perfect-Shuffle Permutations: Definition

The following definition and properties follow the presentation in (Van Loan, 2000).

**Definition C.1** ($(p, q)$ perfect-shuffle permutation matrix (Van Loan, 2000)). Let $p, q \in \mathbb{N}$ and set $r = pq$. The $(p, q)$ perfect-shuffle permutation matrix $\mathbf{P}_{p,q}$ is the $r \times r$ permutation obtained by reshaping a length-$r$ vector into a $p \times q$ matrix in row-major order and then reading it column-wise. Equivalently,

$$\mathbf{P}_{p,q} := \begin{pmatrix} \mathbf{I}_r[R_0, :] \\ \mathbf{I}_r[R_1, :] \\ \vdots \\ \mathbf{I}_r[R_{q-1}, :] \end{pmatrix} \tag{6}$$

where $R_i := \{i + qj \mid j \in [\![0, p-1]\!]\}$ for $i \in [\![0, q-1]\!]$ and $\mathbf{I}_r[R_i, :] \in \{0,1\}^{p \times r}$ is the restriction of the identity matrix $\mathbf{I}_r$ to the rows indexed in $R_i$.

Because the sets $R_0, \ldots, R_{q-1}$ partition $\{0, \ldots, r-1\}$, $\mathbf{P}_{p,q}$ is a (row) permutation of $\mathbf{I}_r$, and in particular is itself a permutation matrix.

### C.2. Block-Diagonalizing a KS Matrix with Perfect-Shuffle Permutations

We now show that $\tilde{\mathbf{K}}$ as defined in Algorithm 1

$$\tilde{\mathbf{K}} := \mathbf{P}^\top \mathbf{K} \mathbf{Q}^\top$$

is block-diagonal when $\mathbf{P}, \mathbf{Q}$ are chosen as perfect-shuffle permutations as in (5) and $\mathbf{K}$ is a KS matrix with pattern $\boldsymbol{\pi} = (a, b, c, d)$. Consider the KS pattern $\tilde{\boldsymbol{\pi}} = (ad, b, c, 1)$, whose support $\mathbf{S}_{\tilde{\boldsymbol{\pi}}} = \mathbf{I}_{ad} \otimes \mathbf{1}_{b \times c}$ corresponds to $ad$ dense diagonal blocks of size $b \times c$. It suffices to establish

$$\mathbf{S}_{\boldsymbol{\pi}} = \underbrace{\big(\mathbf{I}_a \otimes \mathbf{P}_{b,d}\big)}_{:=\mathbf{P}} \mathbf{S}_{\tilde{\boldsymbol{\pi}}} \underbrace{\big(\mathbf{I}_a \otimes \mathbf{P}_{c,d}\big)^\top}_{:=\mathbf{Q}} = \mathbf{P} \, \mathbf{S}_{\tilde{\boldsymbol{\pi}}} \, \mathbf{Q}. \tag{7}$$

Because permutation matrices are orthogonal, $\mathbf{P}^\top \mathbf{S}_{\boldsymbol{\pi}} \mathbf{Q}^\top = \mathbf{S}_{\tilde{\boldsymbol{\pi}}}$. For any compatible matrices $\mathbf{A}, \mathbf{B}, \mathbf{C}$ we have

$$\mathrm{supp}(\mathbf{ABC}) \subseteq \mathrm{supp}\big(\mathrm{supp}(\mathbf{A})\,\mathrm{supp}(\mathbf{B})\,\mathrm{supp}(\mathbf{C})\big),$$

where $\mathrm{supp}(\cdot)$ is the binary support mask. Since $\mathrm{supp}(\mathbf{P}) = \mathbf{P}$ and $\mathrm{supp}(\mathbf{Q}) = \mathbf{Q}$,

$$\mathrm{supp}\big(\mathbf{P}^\top \mathbf{K} \mathbf{Q}^\top\big) \subseteq \mathrm{supp}\big(\mathbf{S}_{\tilde{\boldsymbol{\pi}}}\big) = \mathbf{I}_{ad} \otimes \mathbf{1}_{b \times c},$$

which is the support of a block-diagonal matrix with $ad$ dense $b \times c$ blocks, proving the claim.

*Proof of* (7). A key identity from (Van Loan, 2000) states that for all positive integers $b, c, d$,

$$\mathbf{P}_{b,d}{}^\top \left(\mathbf{1}_{b \times c} \otimes \mathbf{I}_d\right) \mathbf{P}_{c,d} = \mathbf{I}_d \otimes \mathbf{1}_{b \times c}.$$

Because $\mathbf{S}_{\boldsymbol{\pi}} = \mathbf{I}_a \otimes \mathbf{1}_{b \times c} \otimes \mathbf{I}_d$, we obtain

$$\mathbf{S}_{\boldsymbol{\pi}} = \mathbf{I}_a \otimes \big(\mathbf{P}_{b,d}\,(\mathbf{I}_d \otimes \mathbf{1}_{b \times c})\,\mathbf{P}_{c,d}{}^\top\big).$$

Using $(\mathbf{AB}) \otimes (\mathbf{CD}) = (\mathbf{A} \otimes \mathbf{C})(\mathbf{B} \otimes \mathbf{D})$ for any matrices $\mathbf{A}, \mathbf{B}, \mathbf{C}, \mathbf{D}$ of compatible sizes, we factor:

$$\begin{aligned}
\mathbf{S}_{\boldsymbol{\pi}} &= \mathbf{I}_a \otimes \left(\mathbf{P}_{b,d}(\mathbf{I}_d \otimes \mathbf{1}_{b \times c})\mathbf{P}_{c,d}{}^\top\right) \\
&= (\mathbf{I}_a \otimes \mathbf{P}_{b,d}) \left(\mathbf{I}_a \otimes \left((\mathbf{I}_d \otimes \mathbf{1}_{b \times c})\mathbf{P}_{c,d}{}^\top\right)\right) \\
&= (\mathbf{I}_a \otimes \mathbf{P}_{b,d})(\mathbf{I}_a \otimes \mathbf{I}_d \otimes \mathbf{1}_{b \times c})(\mathbf{I}_a \otimes \mathbf{P}_{c,d}{}^\top) \\
&= (\mathbf{I}_a \otimes \mathbf{P}_{b,d})(\mathbf{I}_{ad} \otimes \mathbf{1}_{b \times c})(\mathbf{I}_a \otimes \mathbf{P}_{c,d}{}^\top).
\end{aligned}$$

which is exactly (7). $\qquad\square$

**Algorithmic Implications.** Because $\tilde{\mathbf{K}}$ is block-diagonal, Step (2) of Algorithm 1 can leverage optimized dense kernels on each sub-block. Both BMM and BSR follow this strategy, as summarized in Table 1. The cost, however, is two full-tensor permutations (Steps 1 and 3), which we measure to be a non-negligible share of runtime. Section 4 introduces a *mathematically equivalent* approach that eliminates these permutations altogether.

## D. Software Integration and Implementation Details of the Fused KS Kernel

This appendix complements Section 4 with practical information about the released software, the `lazylinop` integration, and the design choices that guided our CUDA kernel.

### D.1. Integration with `lazylinop`

We provide two complementary entry points for working with Kronecker-sparse matrix multiplications (KSMM), each designed for a specific audience and purpose.

- The GitHub repository `ksmm` is targeted at the machine learning community. It includes all CUDA/OpenCL source code, benchmarks, and a plug-and-play PyTorch module, `KSLinear`, that can replace standard `Linear` layers in neural networks.

- Separately, we are also integrated our fast matrix factorization routines into the `lazylinop` ecosystem, a package more oriented toward signal processing and fast linear transforms. These two entry points are intended to remain independent moving forward.

`lazylinop`[15] is a Python library that supports delayed execution: it builds an operator expression graph lazily and only evaluates it when a final result is requested (delayed evaluation paradigm). One of its key features with respect to

---

[15]https://faustgrp.gitlabpages.inria.fr/lazylinop/index.html

Kronecker-sparsity is the ability to approximate a dense matrix $\mathbf{W}$ into a fast transform, i.e., a product of structured KS factors:

$$\mathbf{W} \approx \mathbf{K}_1 \cdots \mathbf{K}_L$$

This results in fast-transform approximations that might be more efficient in time and energy, especially on parallel hardware.

CUDA and OpenCL backends enable efficient evaluation of these factor chains $\mathbf{K}_1 \cdots \mathbf{K}_L X$ in a *batch-size-last* layout (batch dimension last). The Butterfly module of `lazylinop` supports execution on both CPUs and NVIDIA/AMD GPUs. The documentation is available at https://faustgrp.gitlabpages.inria.fr/lazylinop/index.html and the low-level source code is available at https://gitlab.inria.fr/faustgrp/lazylinop.

The current `lazylinop` package offers the following features for Butterfly and Kronecker-sparse operators:

1) **Butterfly factorization (products of KS matrices) (Le et al., 2024).** Given a dense matrix $\mathbf{W}$, `lazylinop` approximates it as $\mathbf{W} \approx \mathbf{K}_1 \cdots \mathbf{K}_L$ using a quasi-optimal factorization algorithm that achieves the theoretical error bounds of Le et al., all in $\mathcal{O}(MN)$ time for an $M \times N$ matrix.

2) **Compatibility with PyTorch tensors (in progress).** PyTorch tensor inputs and outputs will soon be supported directly within `lazylinop`, streamlining integration into PyTorch-based workflows.

### D.2. Why an Output-Stationary Dataflow?

Like CUTLASS dense kernels (NVIDIA, 2023a) and recent sparse 3-D-convolution kernels (Tang et al., 2023; Spconv Contributors, 2022; Yan et al., 2018), our fused KS kernel is *output-stationary*. Two considerations motivate this choice:

1) *Weight-stationary seems to offer little reuse.* In a KS pattern $(a, b, c, d)$, when the parameter $a$ is large, the matrix is partitioned into $a$ submatrices that act on disjoint regions (Figure 2). Thus, weights are not reused across multiple input/output regions, limiting the benefits of keeping them stationary.

2) *Input-stationary seems harder to parallelize.* Due to the non-contiguous memory accesses involved (Figure 3), read and write operations on inputs/outputs come at a higher cost, so it is preferable to keep one of them stationary to reduce these costs. Both costs are largely driven by the parameter $d$ of KS patterns $(a, b, c, d)$, as it determines the distance between consecutive data elements that should be loaded together when considering one of the dense subproblems (Section 4.1). Since their reuse costs are similar, we had to consider other factors. Input-stationarity poses parallelization challenges as different thread blocks cannot accumulate into the same output coefficient (no possible synchronization) (NVIDIA, 2023a). In contrast, output-stationarity avoids this issue, hence our choice.

### D.3. Kernel-Level Optimizations

Our implementation follows standard high-performance GPU practices:

- **Vectorization.** Wherever possible we use `float4` and `half2` loads/stores to coalesce memory traffic (NVIDIA, 2023b; 2024; 2023a; Boehm, 2022).

- **Double buffering.** Each thread block overlaps computation on the current tile with prefetching of the next tile into shared memory (Li et al., 2019; NVIDIA, 2023b).

- **Structured epilogue.** Partial results are first written to shared memory and then flushed to global memory in a contiguous pattern, avoiding scattered global writes (NVIDIA, 2023a).

Kernel launch parameters (block size, tile shape, vector width) are auto-tuned per pattern $\boldsymbol{\pi} = (a, b, c, d)$ and per GPU architecture, mirroring the tuning strategy of CUTLASS.

## E. Experiments

### E.1. Details on the Experiments

The `pytorch` package version is 2.2 and `pytorch-cuda` is 12.1.

**Matrix Sizes.** In all our experiments with matrices, we set the batch size to $B = 25088$ (196 tokens $\times$ 128 sequences), a standard choice for ViTs as it corresponds to the standard number of tokens per sequence (196) multiplied by the standard number of sequences in a batch of inputs (128). When dealing with a batch of images in neural networks, we choose the standard choice of batch size $B = 128$.

**Matrix Entries.** The nonzero entries of any Kronecker-sparse matrix $\mathbf{K} \in \mathbb{R}^{abd \times acd}$ with sparsity pattern $(a, b, c, d)$ are drawn i.i.d. uniformly in $[-\frac{1}{\sqrt{c}}, \frac{1}{\sqrt{c}}]$, corresponding to the initialization used for training in Dao et al. (2022b). The entries of the inputs $\mathbf{X}$ are drawn i.i.d. according to a standard normal distribution $\mathcal{N}(0, 1)$.

**Benchmarking Time Execution.** All the experiments measuring time execution of a Kronecker-sparse matrix multiplication algorithm (Tables 2, 5, 6, 9 and 10, Figures 4, 6a, 9, 10 and 12 to 18) are performed on a NVIDIA A100-PCIE-40GB GPU associated with an Intel(R) Xeon(R) Silver 4215R CPU @ 3.20GHz with 377G of memory. The full benchmark took approximately 3 days in an isolated environment, ensuring that no other processes were running concurrently.

Measurements are done using the PyTorch tool `torch.utils.benchmark.Timer`. The medians are computed on at least 10 measurements of 10 runs. In $94.2\,\%$ of the cases, we have an interquartile range (IQR) that is at least 100 times smaller than the median (resp. $98\,\%$ for 50 times smaller, and $99.7\,\%$ for 10 times smaller).

**Benchmarking Energy Consumption.** Measurements of the energy consumption (Figure 6b) is done on a NVIDIA Tesla V100-PCIE-16GB GPU associated with an Intel(R) Xeon(R) Silver 4215R CPU @ 3.20GHz with 754G of memory. The full benchmark took approximately 1.5 days in an isolated environment. Measurements are made using the pyJoules software toolkit. The medians are computed on 10 measurements of at least 16 runs. In $96\,\%$ of the cases, the IQR is at least 10 times smaller than the median, and 5 times smaller in all the cases.

**Kronecker-Sparsity Patterns Benchmarked for Time Measurements (Section 5).** The considered patterns are generated by the Python code written in Figure 7. In all the cases, we only consider patterns $(a, b, c, d)$ with $b = c$ or $b = 4c$ or $c = 4b$ to have an input size $N$ and an output size $M$ such that $N = M$ or $N = 4M$ or $M = 4N$. We make this choice because fully-connected layers in Transformers usually satisfy these dimension constraints.

The first "for" loop in Figure 7 (line 21 to 24) generates a wide range of patterns $(a, b, c, d)$ with $a = 1$, as this represents the simplest scenario. Indeed, the case $a > 1$ simply corresponds to repeating $a$ times the case $a = 1$ in parallel.

The second "for" loop in Figure 7 (line 26 to 30) generates patterns with $a > 1$ offering fewer choices for $d$ to keep the benchmark concise in terms of execution time. This loop also imposes additional conditions on $b$ and $c$ (line 28 of the code) that we now explain. Many graphs are plotted based on the ratio $(b + c)/bc$, as introduced in Equation (4). Because of that, our goal was to include as many distinct ratios $(b + c)/bc$ as possible while keeping the benchmark brief. We excluded certain $(b, c)$ values because they resulted in a ratio that was very close to one already in the benchmark and were more computationally intensive.

**Patterns Benchmarked for Energy Measurements (Section 5).** For the energy measurements, the goal is to have diverse sparsity patterns $(a, b, c, d)$ corresponding to many different ratios $d(b + c)/bc$ to observe the trend in Figure 6b, while keeping the benchmark as short as possible. We chose to consider the cartesian product of

```
a_list = [1, 4, 16, 32, 64]
b_list = [48, 64, 96, 128, 192, 256, 384, 512, 768, 1024]
c_list = [48, 64, 96, 128, 192, 256, 384, 512, 768, 1024]
d_list = [1, 2, 3, 4, 6, 8, 12, 16, 24, 32, 48, 64]
```

by skipping as in Figure 7 all the patterns with

```
(b,c) in [(1024 , 256) , (256 , 1024) , (128 , 512) , (512 , 128) , (64 , 256) , (256 , 64)]
```

and also all the patterns such that

```
b != c and b != 4 * c and c != 4 * b
```

for the same reasons as explained above for time measurements.

**Details on boxplots.** In all boxplots (Figures 4, 6a to 6b, 10 and 12 to 18), the orange line corresponds to the median, the

```
1   import itertools
2
3   batch_size = 25_088
4   size_limit = 2_147_483_647
5
6   a_list = [1, 2, 3, 4, 6, 8, 12, 16, 24, 32, 48, 64, 96, 128]
7   b_list = [48, 64, 96, 128, 192, 256, 384, 512, 768, 1024]
8   c_list = [48, 64, 96, 128, 192, 256, 384, 512, 768, 1024]
9   d_list1 = [1, 2, 3, 4, 6, 8, 12, 16, 24, 32, 48, 64, 96, 128]
10  d_list2 = [4, 16, 64]
11
12  def get_patterns_benchmark():
13      patterns_list = []
14
15      def add_pattern(a, b, c, d):
16          if batch_size * a * c * d <= size_limit and \
17              batch_size * a * b * d <= size_limit and \
18              a * b * c * d <= size_limit:
19               patterns_list.append((a, b, c, d))
20
21      for b, c, d in itertools.product(b_list, c_list, d_list1):
22          a = 1
23          if (b == c or b == 4 * c or c == 4 * b):
24              add_pattern(a, b, c, d)
25
26      for a, b, c, d in itertools.product(a_list, b_list, c_list, d_list2):
27          if a != 1 and \
28              (b, c) not in [(1024, 256), (256, 1024), (128, 512), (512, 128), (64, 256),
29                  (256, 64)] and \
                (b == c or b == 4 * c or c == 4 * b):
30               add_pattern(a, b, c, d)
31
32      return patterns_list
```

*Figure 7.* Python code to generate the patterns benchmarked for the execution time in the numerical experiments of Section 5.

boxes to the first and third quartile and the whiskers to the 5th and the 95th percentile. Outliers are not represented on the graph.

### E.2. Performance Across Sparsity Levels (Section 5)

Figure 8 illustrates the distribution of sparsity levels across the 600 KS patterns used in our benchmark. A clear skew toward highly sparse patterns is observed: the *median* sparsity is $97.9\,\%$, and $75\,\%$ of all patterns have a sparsity of at least $91.7\,\%$. As a result, any aggregated statistic computed over the full set of 600 patterns—such as the medians reported in Table 2—tends to be dominated by these very sparse cases.

To assess whether our findings hold across the entire sparsity spectrum, we complement the overall results in Table 2 with a stratified analysis by sparsity level. Table 5 reports the performance of KERNEL compared to three existing KS-aware baselines—BMM, EINSUM, and BSR—within six sparsity bins. These bins were defined by equally spaced quantiles in log-sparsity scale, to reflect the long-tailed distribution of sparsity levels.

Table 5 confirms that our fused KERNEL consistently outperforms the baselines in each of the different sparsity regimes. Furthermore, within each bin, we compute the correlation between the speedup of KERNEL and the heuristic $h(b,c) = \frac{b+c}{bc}$. In all six bins, we observe strong positive correlations (ranging from $0.76$ to $0.91$ in log-log scale), indicating that $h(b,c)$ remains a strong predictor of performance independently of sparsity. This supports the conclusion that our kernel's speedup is driven primarily by the structural pattern captured by $h(b,c)$, rather than by the density of nonzero entries.

We also provide a visual counterpart in Figure 9, which extends the analysis of Figure 6a by showing the variation in speedup as a function of $h(b,c)$ at *fixed* sparsity levels, rather than aggregated bins. This further illustrates the dominant role of the heuristic in explaining performance gains.

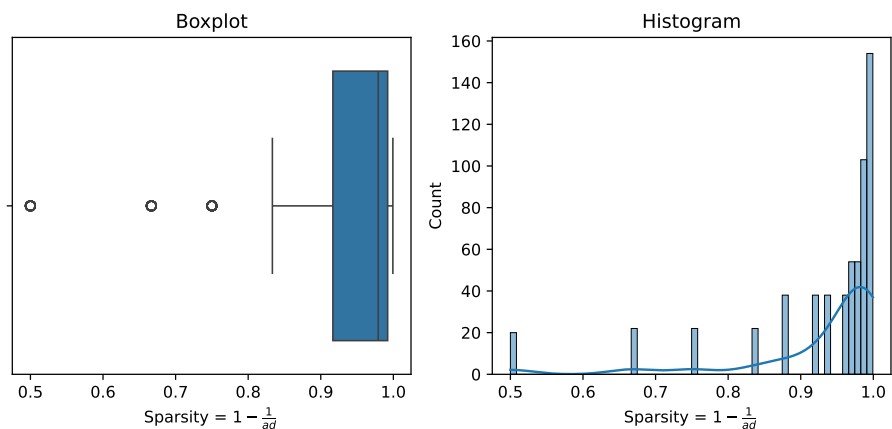

*Figure 8.* Boxplot and histogram of the sparsity levels covered by the 600 KS patterns of our benchmark in Section 5.

*Table 5.* Win rate and median speed-up of KERNEL vs $\min\{\text{BMM}, \text{EINSUM}, \text{BSR}\}$ across sparsity bins, in FP32. The correlation is between $\log(\text{speedup})$ and $\log(h)$ within each bin, where $h(b, c) = \frac{b+c}{bc}$.

| Sparsity bin | Win rate | Median × faster | Number of patterns | Corr(log-log) |
|---|---|---|---|---|
| 99.48–99.93% | 100.0% | 1.84 | 102 | 0.91 |
| 98.96–99.48% | 93.1% | 1.41 | 101 | 0.82 |
| 97.92–98.96% | 79.6% | 1.38 | 108 | 0.83 |
| 95.83–97.92% | 81.5% | 1.41 | 92 | 0.80 |
| 87.49–95.83% | 89.5% | 1.30 | 114 | 0.80 |
| 49.99–87.49% | 86.0% | 1.23 | 86 | 0.76 |

### E.3. Measuring the Cost of the Permutation Steps in BMM (Section 3)

**Protocol.** The BMM baseline follows Algorithm 1, which for a pattern $(a, b, c, d)$ performs *two* global permutations (lines 1 and 3) plus one matrix multiplication with the permuted factor $\tilde{\mathbf{K}}$. Because $\tilde{\mathbf{K}}$ itself has pattern $(ad, b, c, 1)$, we isolate the permutation overhead by timing:

1. $\Delta t$: the full BMM kernel on the original pattern $(a, b, c, d)$;
2. $\Delta \tilde{t}$: the same kernel restricted to the multiplication step only, i.e., BMM executed on $(ad, b, c, 1)$.

The share of runtime devoted to the two permutations is

$$\frac{\Delta t - \Delta \tilde{t}}{\Delta t}.$$

**Results.** Figure 10 shows the distribution of memory transfer overheads at a fixed batch size of 25088. The patterns included in this plot are of the form $(1, b, c, d)$ with $b, c \in \{48, 64, 96, 128, 192, 256, 384, 512, 768, 1024\}$ such that $b = c$, $b = 4c$, or $c = 4b$, and $d \in \{1, 2, 3, 4, 6, 8, 12, 16, 24, 32, 48, 64, 96, 128\}$.

In the *batch-size-first* layout (Figure 10a, reproduced from Figure 4), the overhead of BMM due to the two global permutations in Algorithm 1 increases with the ratio $h(b, c) = \frac{b+c}{bc}$, reaching up to 50 %.

The same trend is observed in the *batch-size-last* layout (Figure 10b).

### E.4. Details on min time(KERNEL, BMM, BSR, EINSUM) vs. min time(DENSE, SPARSE) (Section 5.4)

Figure 12 shows that the speedup factor of implementations specialized to the Kronecker-sparsity (KERNEL, BMM, BSR, EINSUM) over the generic DENSE and SPARSE implementations increases with the matrix size $M \times N$. Figure 11 describes the distribution in sizes (the product $MN$) over the benchmark: most of the values of $MN$ are in between $4.2 \times 10^6$ (first quartile) and $6.0 \times 10^8$ (third quartile). We recall that the output and input dimensions are $M = abd$ and $N = acd$,

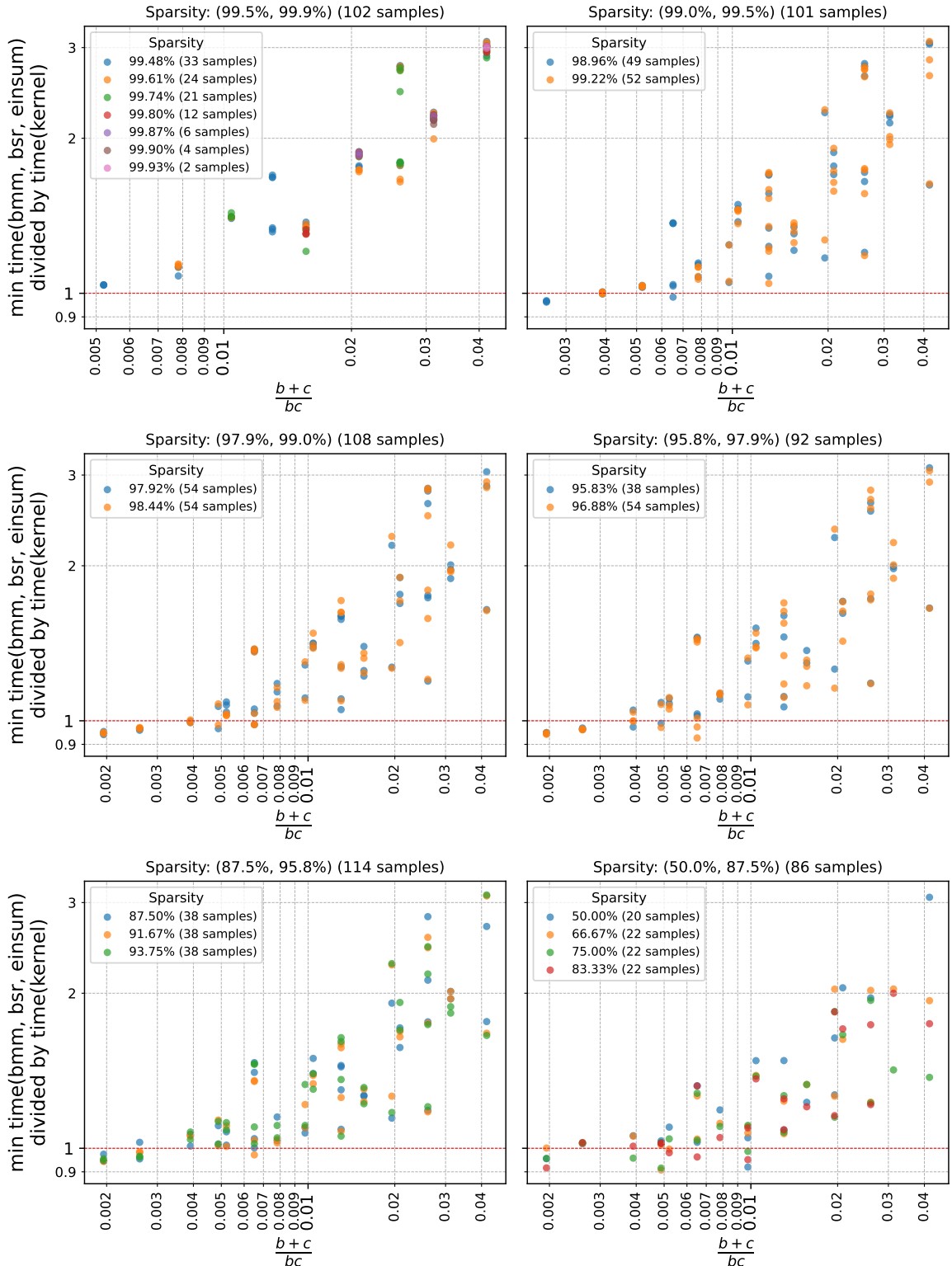

*Figure 9.* Speedup factor of KERNEL compared to baselines (BMM, BSR, EINSUM) vs. the heuristic $h(b, c) = \frac{b+c}{bc}$, when fixing the sparsity level at a given value, in FP32. Each subplot corresponds to different sparsity bins involved in Table 5.

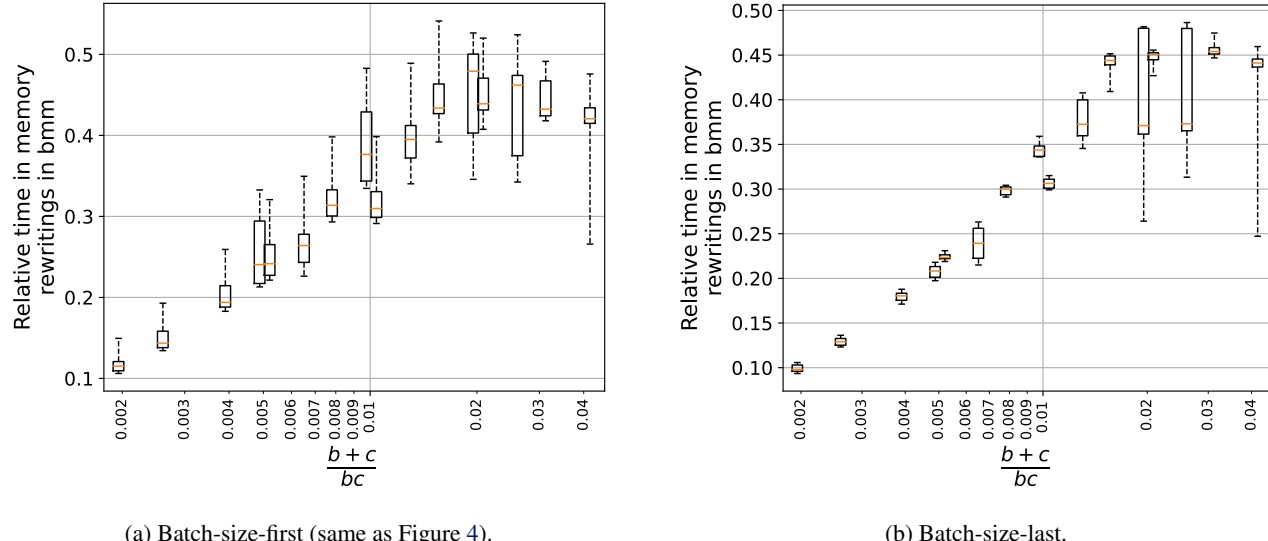

(a) Batch-size-first (same as Figure 4).

(b) Batch-size-last.

*Figure 10.* Estimated share of runtime devoted to the permutation steps in the BMM baseline for many patterns $\boldsymbol{\pi} = (a, b, c, d)$, in FP32. Patterns are grouped by the value of $h(b, c) = \frac{b+c}{bc}$ and the distribution within each group is shown as a boxplot.

respectively, for a Kronecker-sparse matrix with pattern $\boldsymbol{\pi} = (a, b, c, d)$.

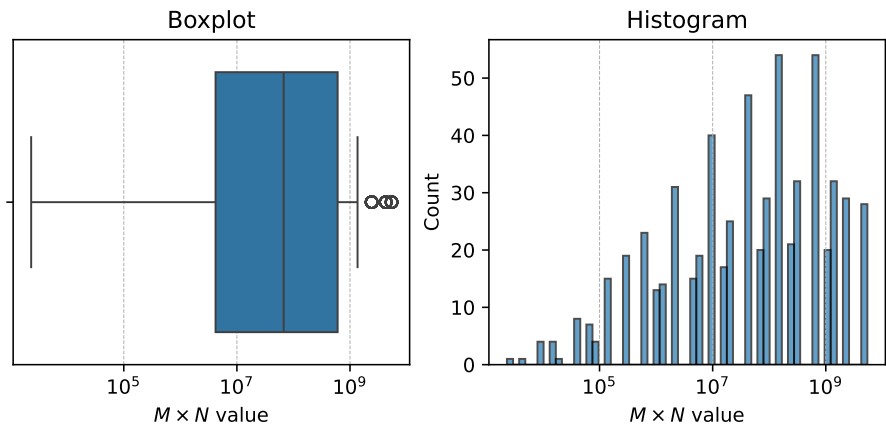

*Figure 11.* Distribution of matrix sizes in the benchmark.

### E.5. Details on time(BMM) vs. min time(BSR, EINSUM) (Section 5.4)

Figure 13 shows that for a sufficient large matrix size $M \times N$, we always have time(BMM) $<$ min time(BSR, EINSUM), i.e., the BMM implementation is the most efficient among all baseline implementations (BMM, EINSUM, BSR).

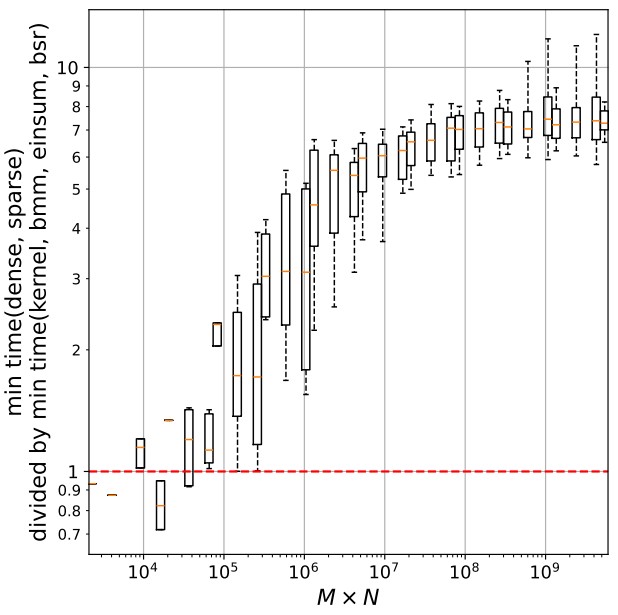

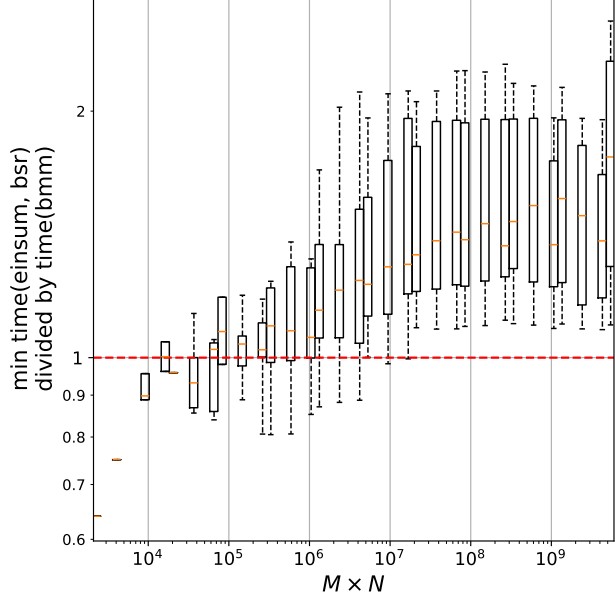

*Figure 12.* Speedup factor of min time(KERNEL, BMM, BSR, EIN-SUM) compared to min time(DENSE, SPARSE) as a function of the matrix size $M \times N$.

*Figure 13.* Speedup factor of time(BMM) compared to min time(EINSUM, BSR) as a function of the matrix size $M \times N$.

### E.6. Details on the Impact of the Memory Layout (Section 5.4)

Figure 14a shows how the memory layout (*batch-size-first* vs. *batch-size-last*) impacts the runtime of each implementation. In particular, it shows that KERNEL achieves a median speedup of approximately ×2 when switching from *batch-size-first* to *batch-size-last*.

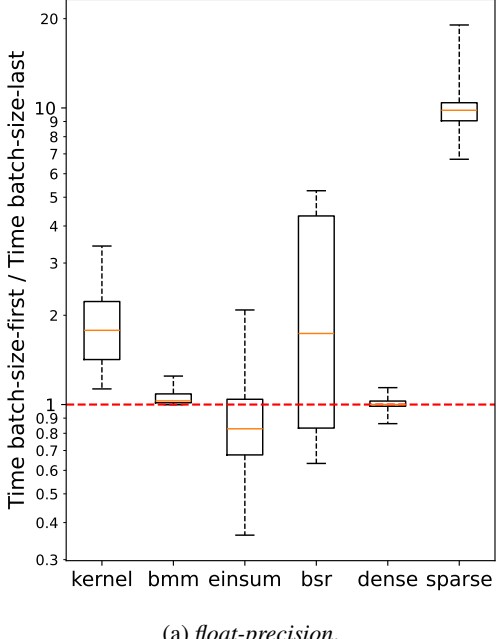

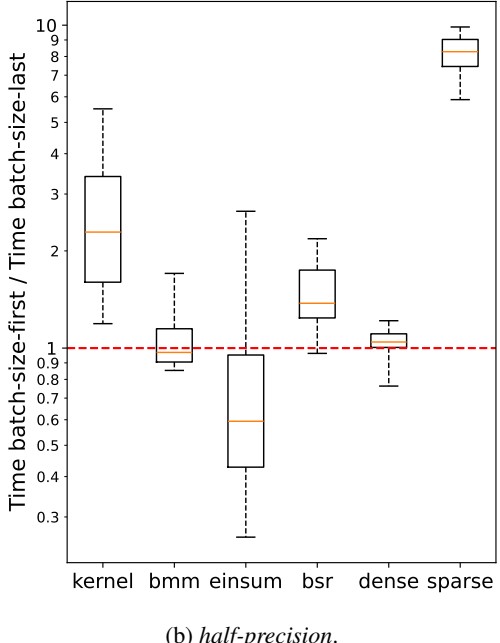

(a) *float-precision.*

(b) *half-precision.*

*Figure 14.* Boxplots of the ratio $\frac{\text{time of } batch\text{-}size\text{-}first}{\text{time of } batch\text{-}size\text{-}last}$, in FP32.

Table 6 reports the percentage of KS patterns for which KERNEL outperforms all baseline implementations, in either the *batch-size-first* or *batch-size-last* memory layout. Even when all implementations are restricted to the *batch-size-first* layout, KERNEL still improves over all baselines on $20\%$ of the tested patterns, despite relying on some non-contiguous memory accesses (Section 4.2).

Table 6 also confirms that among existing baselines (excluding the new KERNEL), BMM is the fastest across both layouts, outperforming EINSUM, BSR, DENSE, and SPARSE.

*Table 6.* Percentage out of 600 patterns $(a, b, c, d)$ where algo1 is *faster* than the algo2 (denoted by time(algo1) < time(algo2)), and the median acceleration factor in such cases (that is, the median ratio $\frac{\text{time of algo2}}{\text{time of algo1}}$ for a given memory layout), in FP32.

| | Batch-size-first | | Batch-size-last | |
|---|---|---|---|---|
| Comparison | Win rate | Median × faster | Win rate | Median × faster |
| min{KERNEL, BMM, EINSUM, BSR} < min{DENSE, SPARSE} | 97.61% | ×18.50 | 98.09% | ×6.53 |
| BMM < min{EINSUM, BSR, DENSE, SPARSE} | 94.1% | ×1.44 | 89.47% | ×1.66 |
| KERNEL < min{BMM, EINSUM, BSR, DENSE, SPARSE} | 20.41% | ×1.26 | 85.49% | ×1.39 |

## E.7. Time Spent in Linear Layers in Vision Transformers (Section 6)

This section provides a numerical lower bound estimate of the time spent in fully-connected layers within Vision Transformers (ViTs).

**Results.** Table 7 shows the proportion of computation time dedicated solely to the linear layers in the feed-forward network (FFN) modules across various ViT architectures. This proportion ranges from $31\%$ to $53\%$ in *half-precision* and from $46\%$ to $61\%$ in *float-precision*. The fraction increases with model size, indicating that a significant part of ViT inference is spent in fully-connected layers. Note that our measurement excludes the linear layers in the multi-head attention modules, so these values represent a *lower bound* on the actual time spent in *all* linear layers of the Transformer.

*Table 7.* Median execution times (ms) for the forward pass in various ViTs, and for an MLP composed solely of the linear layers from the feed-forward network modules of the ViTs. The table also reports the proportion of the latter over the former. FP16 corresponds to *half-precision*, and FP32 to *float-precision*.

| ARCHITECTURE | FP16 (S) | | FP32 (S) | |
|---|---|---|---|---|
| | COMPLETE | LINEAR IN FFNS | COMPLETE | LINEAR IN FFNS |
| VIT-S/16 | 0.014 | 0.0046 (31%) | 0.090 | 0.04 (46%) |
| VIT-B/16 | 0.036 | 0.015 (42%) | 0.30 | 0.16 (54%) |
| VIT-L/16 | 0.11 | 0.050 (46%) | 1.0 | 0.58 (58%) |
| VIT-H/14 | 0.31 | 0.16 (53%) | 2.6 | 1.6 (61%) |

**Details on the Estimation.** A Transformer architecture consists of a sequence of blocks, each containing a multi-head attention module and a feed-forward network module. The feed-forward network (FFN) is a two-layer MLP (without biases) with an intermediate non-linear activation. Table 7 reports the cumulative execution time of all linear layers in the FFNs, computed sequentially. This is then compared with the total forward pass time of the full ViT model. Since we do not include linear layers from the attention modules, the reported ratios represent only a lower bound on the total time spent in linear operations.

**Experimental Settings.** The ViT-S/16 architecture follows (Zhai et al., 2022), while ViT-B/16, ViT-L/16, and ViT-H/14 follow (Dosovitskiy et al., 2020). Input images are of size $224 \times 224$. For *float-precision*, we use the PyTorch ViT implementations of Wang (2024b). For *half-precision*, the Transformer uses FlashAttention (Dao et al., 2022c) to compute scaled dot-product attention, as in (Wang, 2024a). The FFN-only MLPs are implemented using `torch.nn.Sequential` and `torch.nn.Linear`. All experiments were run on a single NVIDIA A100-40GB GPU paired with an AMD EPYC 7742 64-Core Processor. Execution times were measured using `torch.utils.benchmark.Timer`, with a batch size of 128.

**E.8. Details on End-to-End Transformers Inference Acceleration (Section 6)**

**Why Do We Focus More on Transformers Rather Than, e.g., Convolutional Neural Networks?** In Section 6, we argued that injecting KS in linear layers is straightforward—making architectures like Transformers a natural fit.

By contrast, applying KS to convolutional neural networks is more challenging. There are at least two ways to inject Kronecker sparsity in convolutional layers, none of which we found convincing enough to be considered in this paper.

The first approach—replacing the convolution kernel with a Kronecker-sparse matrix—is limited by the small size of typical convolution kernels (e.g., $7 \times 7$), leaving little room for meaningful gains.

The second approach—recasting convolutions as matrix multiplications with butterfly-structured weights, as in (Lin et al., 2021)—introduces significant overhead in the form of input/output folding and unfolding operations. In our view, this makes the approach impractical for efficient inference.

For these reasons, we focus on fully connected layers, where KS integration is both simple and impactful.

**Selected Kronecker-Sparse Patterns for ViT-S/16.** In our ViT-S/16 experiments, we replace the dense weight matrices in all fully-connected layers—namely, the projection matrices in the self-attention blocks and the linear layers in the feed-forward network blocks—with products of two Kronecker-sparse matrices $\mathbf{K}_1 \mathbf{K}_2$, where each $\mathbf{K}_i$ follows a specific sparsity pattern $\boldsymbol{\pi}_i$ as defined in Definition 2.1. The patterns used are:

- $(\boldsymbol{\pi}_1, \boldsymbol{\pi}_2) = ((1, 192, 48, 2), (2, 48, 192, 1))$ for square matrices of size $N \times N$ (projections in self-attention blocks),

- $(1, 768, 192, 2), (6, 64, 64, 1)$ for $4N \times N$ matrices (UP projection matrices),

- $(6, 64, 256, 1), (1, 128, 128, 3)$ for $N \times 4N$ matrices (DOWN projection matrices).

These patterns were selected based on two criteria: (i) the total number of nonzero entries across both KS factors is at least 75 % of the original dense matrix size ($a_1 b_1 c_1 d_1 + a_2 b_2 c_2 d_2 \geq 0.75MN$), ensuring sufficient layer expressivity to raise hope that it could yield competitive task accuracy compared to the dense implementation, if it were fine-tuned or trained from scratch on a given task; (ii) the patterns yield a high value of the heuristic $h(b, c) = \frac{b+c}{bc}$, which favors the performance of our fused KERNEL implementation.

**Additional Results for ViT-S/16.** Table 8 presents the latency reduction achieved by replacing standard Linear layers with Kronecker-sparse layers using our fused KERNEL implementation, as compared to the BMM baseline. These results, measured on an Nvidia A100 40GB GPU, confirm that the gains observed at the layer level carry over to larger functional blocks in the model.

*Table 8.* Acceleration of submodules of a ViT-S/16 using Kronecker-sparse matrices, in FP32.

| | $\frac{\text{time(BMM)}}{\text{time(fully-connected)}}$ | $\frac{\text{time(KERNEL)}}{\text{time(fully-connected)}}$ |
|---|---|---|
| Linear $N \times N$ | 0.82 | **0.50** |
| Linear $N \times N$ + bias | 0.97 | **0.66** |
| Linear $4N \times N$ | 0.80 | **0.78** |
| Linear $4N \times N$ + bias | 0.93 | **0.90** |
| Linear $N \times 4N$ | 0.91 | **0.58** |
| Linear $N \times 4N$ + bias | 0.94 | **0.61** |
| FFN ($2 \times$ Linear+GELU+LN) | 0.91 | **0.77** |
| MHA (QKV + proj) | 0.87 | **0.79** |
| Transformer block | 0.90 | **0.78** |
| **ViT-S/16 full** | 0.89 | **0.78** |

**Selected Kronecker-Sparse Patterns for GPT-2 Medium.** For GPT-2 Medium, we inject Kronecker sparsity into the down-projection linear layer within the feed-forward block. The layer in question has dimensions $1024 \times 4096$ (hidden size $d = 1024$). The applied KS patterns are:

$$\boldsymbol{\pi}_1 = (1, 64, 256, 16), \qquad \boldsymbol{\pi}_2 = (64, 64, 64, 1),$$

yielding the factorization $\mathbf{W} = \mathbf{K}_1 \mathbf{K}_2$. These patterns are selected following the same rationale as for ViT-S/16: they are expressive enough (nonzero density above 75 %) and yield high values of the heuristic $h(b, c)$, which predicts good performance with KERNEL.

**Inference Setup.** Inference is conducted in PyTorch using the *batch-size-first* memory layout (batch-size-first), which is the framework's default. All operations aside from the KS layers use standard PyTorch kernels. For ViT-S/16, we use an Nvidia A100 GPU with a batch of 128 image token sequences, each containing 196 tokens. For GPT-2 Medium, we use an Nvidia RTX6000 GPU under similar batch settings.

### E.9. Benchmark Results in Half-Precision

We now reproduce the benchmark from Section 5 in *half-precision*. The *half-precision* equivalents of our key *float-precision* results—Tables 2 and 6, Figure 6a, Figure 10, Figure 12, Figure 13, and Figure 14a—are respectively shown in Tables 9 and 10, Figure 18, Figure 17, Figure 15, Figure 16, and Figure 14b.

As in the FP32 benchmark, Figure 18 reports results only for patterns where at least one KS-aware method (KERNEL, BMM, EINSUM, BSR) is faster than the generic baselines (DENSE, SPARSE). This filters out roughly 14.3 % of the patterns and retains 85.7 % of the benchmarked grid in *half-precision* (see Table 9).

Overall, KERNEL shows fewer speedups compared to the FP32 setting, raising two possible interpretations: either existing baselines have been better optimized for half-precision and benefit more from tensor cores, or there remains untapped room for optimization in our kernel for *half-precision*.

1) Figure 15 and Table 10 confirm that, as in FP32, the implementations tailored to KS matrices (KERNEL, BMM, EINSUM, BSR) still largely outperform the generic baselines (DENSE, SPARSE).

2) Figure 16 and Table 10 show that among the existing KS-aware implementations (BMM, EINSUM, BSR), BMM remains the fastest when the matrix size is sufficiently large, regardless of the memory layout.

3) Figure 17 highlights that in *half-precision*, the relative cost of the permutation steps in BMM grows even larger than in *float-precision*—ranging from 40 % to 80 %, compared to 10 % to 50 % in *float-precision*—further motivating efforts to eliminate these costly memory operations.

4) Figure 18 shows that the new KERNEL can still yield up to $\times 2$ speedups, but only for KS patterns where a large portion of the runtime is spent on permutations—those with high values of the heuristic $h(b, c) = \frac{b+c}{bc} \geq 0.02$. This contrasts with the FP32 case, where improvements already appear from $h(b, c) \geq 0.004$ (see Figure 6a). Consequently, KERNEL improves over existing methods on 38 % of the patterns in *half-precision* (Table 9), compared to 85 % in *float-precision* (Table 2).

5) Table 10 shows that the current KERNEL remains competitive in *batch-size-last* under *half-precision*, but falls behind in *batch-size-first*. This shows that memory layout plays a key role in making Kronecker sparsity efficient on GPUs, and suggests further study on how layout affects other neural network operations.

These findings suggest that while KERNEL remains competitive in *half-precision*, its performance gains are currently more limited. This highlights potential for further optimization, particularly to better leverage half-precision hardware features such as tensor cores.

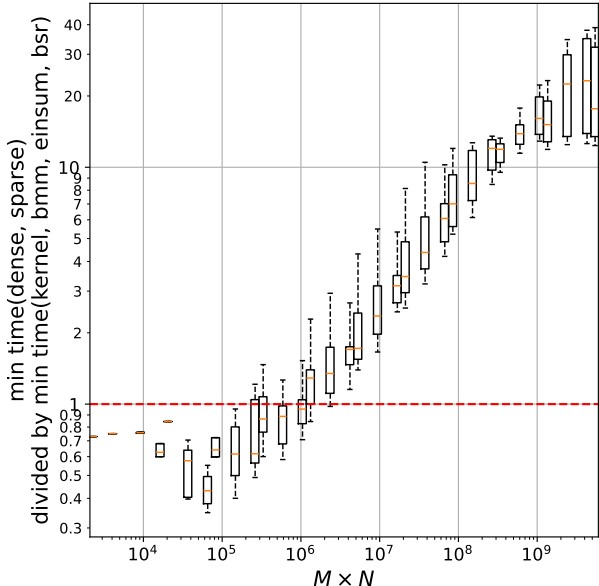

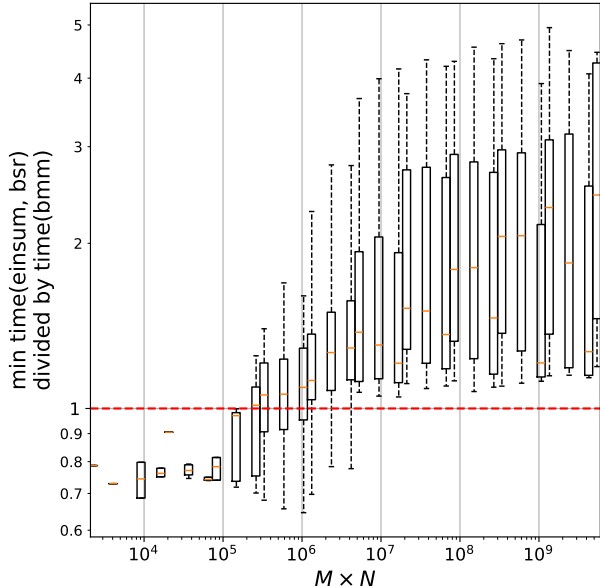

*Figure 15.* Speedup factor of min time(KERNEL, BMM, BSR, EIN-SUM) compared to min time(DENSE, SPARSE) vs. the matrix size $M \times N$. Experiments are carried in *half-precision*.

*Figure 16.* Speedup factor of time(BMM) compared to min time(EINSUM, BSR) vs. the matrix size $M \times N$. Experiments are carried in *half-precision*.

*Table 9.* Percentage out of 600 patterns $(a, b, c, d)$ where `algo1` is *faster* than the `algo2` in *half-precision* (denoted by time(`algo1`) < time(`algo2`)), and the median acceleration factor in such cases (that is, the median ratio $\frac{\text{time of algo2}}{\text{time of algo1}}$). For each implementation, we take the minimum time between the *batch-size-first* and the *batch-size-last* memory layout. Experiments are carried in *half-precision*.

| min time $\begin{pmatrix} \text{KERNEL} \\ \text{BMM} \\ \text{EINSUM} \\ \text{BSR} \end{pmatrix}$ < min time $\begin{pmatrix} \text{DENSE} \\ \text{SPARSE} \end{pmatrix}$ | time(BMM) < min time $\begin{pmatrix} \text{EINSUM} \\ \text{BSR} \\ \text{DENSE} \\ \text{SPARSE} \end{pmatrix}$ | time(KERNEL) < min time $\begin{pmatrix} \text{BMM} \\ \text{EINSUM} \\ \text{BSR} \\ \text{DENSE} \\ \text{SPARSE} \end{pmatrix}$ |
|---|---|---|
| 85.7 % (×8.14) | 80.7 % (×1.52) | 37.8 % (×1.51) |

*Table 10.* Across 600 patterns, this table reports how often each algorithm is the fastest, along with the corresponding median speedup, under each memory layout (*batch-size-first* and *batch-size-last*) in FP16.

| Comparison | *Batch-size-first* | | *Batch-size-last* | |
|---|---|---|---|---|
| | Win rate | Median × faster | Win rate | Median × faster |
| min{KERNEL, BMM, EINSUM, BSR} < min{DENSE, SPARSE} | 83.18 % | ×6.31 | 85.34 % | ×8.14 |
| BMM < min{EINSUM, BSR, DENSE, SPARSE} | 82.25 % | ×1.54 | 79.48 % | ×2.65 |
| KERNEL < min{BMM, EINSUM, BSR, DENSE, SPARSE} | 0.0 % | - | 41.78 % | ×1.56 |

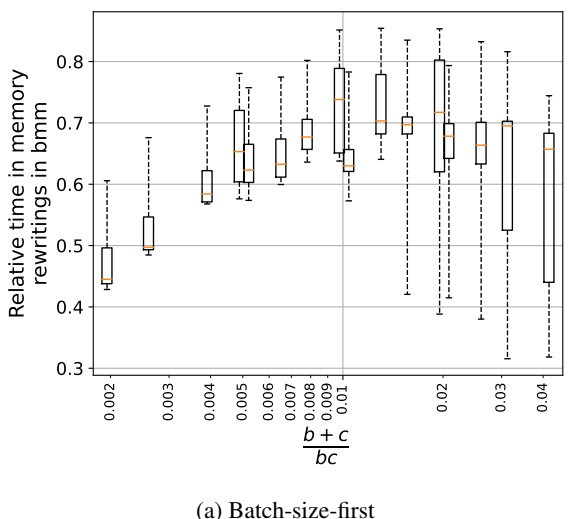

(a) Batch-size-first

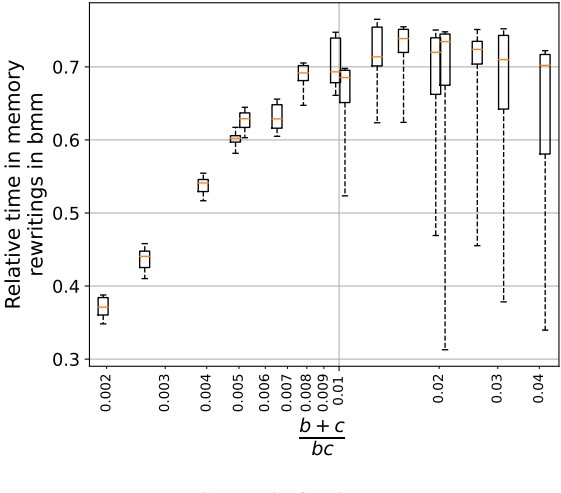

(b) Batch-size-last

*Figure 17.* Estimated relative time spent on memory rewritings in BMM for the multiplication with $\mathbf{K} \in \mathcal{K}_{\boldsymbol{\pi}}$, for several $\boldsymbol{\pi} = (a, b, c, d)$. We regroup patterns by their value of $(b + c)/(bc)$, and plot a boxplot to summarize the corresponding measurements. Experiments are carried in *half-precision*.

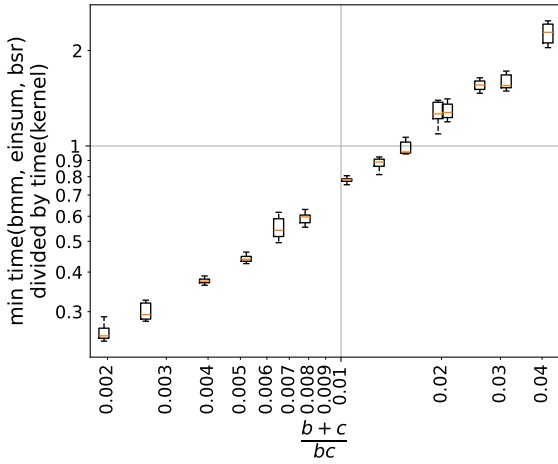

*Figure 18.* Speedup factor of KERNEL compared to $\min(\text{BMM}, \text{EINSUM}, \text{BSR})$ in *half-precision*. For each implementation, we take the minimum time between the *batch-size-first* and the *batch-size-last* memory layout. We regroup the $(a, b, c, d)$ patterns by their value of $(b + c)/(bc)$, and use a boxplot to summarize the corresponding measurements. Experiments are carried in *half-precision*.

