# OpenReview forum: "Fast Inference with Kronecker-Sparse Matrices"
_ICML.cc/2025/Conference — ICML 2025 poster_

### Official Review · Reviewer_XhSi · 2025-03-13

**Overall Recommendation:** 4

**Summary:**

This paper presents the first energy and time benchmarks for the multiplication of Kronecker-sparse matrices. These benchmarks reveal that specialized sparse matrix multiplication implementations spend up to 50% of run time on memory rewrite operations. As a remedy, the authors propose a new tiling strategy for Kronecker-sparse matrix multiplication achieving a median speed-up of 1.4x while also cutting energy consumption by 15%.

**Claims And Evidence:**

That authors appear to be making a few main claims.

1. The first claim is that previous approaches to handling Kronecker-sparse matrix multiplications spend up to 50% of their total runtime on memory rewriting operations. This is supported empirically by a set of results reported by the authors.

2. The second claim is that the time spent on memory rewriting operations can be attributes to the structure adapted by most such algorithms, which can be dropped by a factor of three by tiling, essentially coalescing the read/write operations. The authors have supported this claim analytically.

3. The third claim concerns empirical improvements, where the authors have claimed they new Kronecker-sparse multiplication algorithm to lead to speeds ups of up to 1.4x and energy savings of 15%, as well as improve the efficiency of neural network inference. The authors have support all of these claims empirically.

**Essential References Not Discussed:**

Not aware of any.

**Experimental Designs Or Analyses:**

I found the experimental design and analysis to be sound.

The authors start with a hypothesis that current matrix multiplication algorithms used with Kronecker-sparse matrices spend a significant portion of their run time on memory rewriting, a hypothesis which they empirically validate on 600 different sparsity patterns. They then devised a theoretically-equivalent algorithm with a theoretically lower number of memory rewriting operations. They then validate that they new algorithm is indeed faster than the current used algorithms as well as more energy efficient, due to having to perform less energy on memory rewritings.

**Methods And Evaluation Criteria:**

Yes, the methods and evaluation criteria make sense. The only thing is I was expecting to see comparison to more sparse matrix multiplication baselines but I am not very familiar with the area, so maybe such specialized algorithms do not exist or aren't used in practice?

**Other Comments Or Suggestions:**

Typos: Section 5 326-363 line/row -> column?

**Other Strengths And Weaknesses:**

- I found the paper to be very well written despite the very technical subject matter. I greatly appreciated the top-down manner through which the problem was decomposed, the hypothesis validated and the method proposed to tackle the shortcoming of the current approaches.

- I like how the authors did not stop at comparing their algorithm to previously proposed algorithms in a synthetic setting, but rather also showed that practical settings, such as inference in ViT architectures can benefit greatly in terms of speedup.

- To my knowledge, this contribution is novel, and I foresee it having a sizable impact in the community's effort to improve the efficiency of inference in deep models.

- I was expected to see quite a bit more in terms of related work given that sparsity is currently a highly-researched area

**Questions For Authors:**

The only question that I have is: What is the point of diminishing returns? My understanding is there must be a threshold whereby if the matrix does not exhibit enough sparsity then the baselines might perform better? Am I correct in my assumption? I believe this might also be suggested by the The third column of Table 3 that  shows that kernel is slower than all other baselines 12% of the tested patterns? Any idea why that might be the case?

**Relation To Broader Scientific Literature:**

The contribution put forth in this paper is very timely and application to machine learning community, but I suspect also to the algorithms and scientific computing communities. This work is essentially on how to speed up matrix multiplication when we know of, or impose, a specific sparsity structure on one of our input matrices.

**Theoretical Claims:**

Yes, I went over proofs and theoretical claims and they appear to make sense to me

---

> ### Author Rebuttal · Authors · 2025-03-31
>
> Thank you for your review. We address your points below.
>
> 1. >Are there any other sparse related works relevant for this benchmark?
>
> The benchmark includes all the relevant baselines we are aware of. The revision will include an additional discussion clarifying how our work relates to a few other areas of sparse matrix research (such as sparse 3d convolutions or sparse tensor compilers, as suggested by other reviewers), even though these related works do not offer relevant implementations to be included in the benchmark.
>
> 2. > On the potential existence of a sparsity threshold under which previous baselines are better
>
> You are absolutely right, thank you for mentioning that. The 12% of the cases where the baselines are still better correspond to patterns that have a high density of nonzero or a small value of the proposed heuristic (the ratio (b+c)/bc). The revision will contain a plot that shows how the speedup increases with the sparsity level (percentage of zeros).
>
> We also thank the reviewer for spotting typos.

---

### Official Review · Reviewer_nKiB · 2025-03-13

**Overall Recommendation:** 2

**Summary:**

This paper proposes a novel CUDA kernel designed to accelerate neural network inference using Kronecker-sparse matrices. These matrices, characterized by sparsity patterns derived from the Kronecker product, offer a structured alternative to traditional dense matrices in neural networks. By optimizing memory access and reducing redundant operations, the proposed kernel achieves a 1.4× speedup and a 15% reduction in energy consumption compared to existing approaches, demonstrating its effectiveness in enhancing computational efficiency.

## update after rebuttal
I have read author's response, but I still believe that the conversion cost could be high when the proposed method operates on activations—such as in self-attention operations (e.g., Q @ Kᵀ, Attention-Score @ V)—which was my main concern. Since my concern regarding this point have not been fully resolved.

**Claims And Evidence:**

- The study evaluates existing GPU implementations, identifies inefficiencies in memory access, and proposes a new CUDA kernel optimized for Kronecker-sparse matrix multiplications.
- The paper is well-supported by empirical benchmarks and theoretical analysis, with the 1.4× speedup and 15% energy reduction demonstrated through extensive experiments across various sparsity patterns.
- However, the study could be further strengthened by evaluating its performance on different hardware platforms (e.g., AMD GPUs, CPUs, FPGAs) and offering deeper insights into automated sparsity pattern selection.

**Essential References Not Discussed:**

- None

**Experimental Designs Or Analyses:**

- The benchmarking of execution time and energy consumption on various GPU implementations is relevant and well-justified.
- The comparison with existing PyTorch implementations (bmm, bsr, einsum) and generic dense/sparse approaches provides a meaningful baseline.
- However, there is no direct comparison with other existing Kronecker-sparse matrix calculation techniques.

**Methods And Evaluation Criteria:**

- The paper proposes a new tiling strategy for matrix multiplication with Kronecker-sparse matrices, which reduces memory transfer overhead in GPU computations.
- The study carefully selects a range of Kronecker-sparsity patterns and tests them across different conditions, ensuring that results are not limited to specific cases.
- The evaluation extends to practical scenarios like transformer inference acceleration, demonstrating the broader utility of the proposed method.

**Other Comments Or Suggestions:**

- None

**Other Strengths And Weaknesses:**

Strengths
- The paper introduces a new tiling strategy for GPU matrix multiplication, reducing memory access overhead.
- The proposed CUDA kernel achieves up to 1.4× speedup and 15% energy reduction compared to existing methods.
- Demonstrates that the optimized kernel can accelerate vision-transformer model inference.

Weaknesses
- The paper does not compare its CUDA kernel with other structured matrix optimization kernels (e.g., Monarch matrices, Butterfly Transform).
- The study assumes W (weight matrix) is already in Kronecker-sparse format, but does not discuss the cost of converting a dense matrix to this structure. If the transformation cost is high, the practical benefit of the proposed optimization may be reduced in self-attention operation (ex, Q @ K^T, Attention-Score @ V).
- Only the speed difference by unit operation is shown and the End-to-End model (ViT) latency results are not provided.

**Questions For Authors:**

- Can you provide end-to-end latency comparison results measured in ViT?
- Can you provide a latency comparison with other structured matrix optimization kernels (e.g., Monarch matrices, Butterfly Transform) rather than a comparison with the code implemented in pytorch?

**Relation To Broader Scientific Literature:**

- None

**Theoretical Claims:**

- The paper demonstrates that existing methods incur significant GPU memory access costs.
- It quantifies the memory rearrangement cost as (b + c) / (bc) , arguing that a higher value of this ratio indicates the inefficiency of existing approaches.
- However, the theoretical analysis presented in the paper does not clearly establish whether the proposed method guarantees consistent performance improvements across all Kronecker-sparse patterns.

---

> ### Author Rebuttal · Authors · 2025-03-31
>
> Thank you for your review.
>
> # Regarding your questions
>
> 1. > End-to-end latency results in ViT
>
> Table 4 already provides an end-to-end latency result, showing a 22% relative time gain on a vision transformer when using the kernel. If you actually meant to ask about the *absolute* measurements in seconds rather than the *relative* comparison, we would be happy to add them upon request.
>
> 2. > Comparison to other “existing Kronecker-sparse matrix calculation techniques” / “structured matrix optimization kernels” such as Monarch Transform or Butterfly Transform
>
> The implementation associated with the Monarch Transform [1,2] corresponds to the “bmm” baseline in the benchmark. The official code associated with the Butterfly Transform [3,4] is no longer maintained and we were unable to make it work. However, we tested a faithful reproduction internally but found it significantly slower than the other baselines, so we chose not to include it in the benchmark. The revision will make this explicit.
>
> # Regarding the other points you mentioned
>
> 3. > Further strengthen the study by extending it to other hardwares
>
> We agree that exploring the opportunities and challenges related to benchmarking and optimising the kernel on other hardwares is an exciting open avenue that is now raised by this work. To support further exploration, we will release an OpenCL version of the kernel, enabling users with specific requirements to test it on platforms such as AMD GPUs or CPUs.
>
> 4. > CUDA kernel code not included
>
> The final version will include a template of the kernel code and a link to the open-source (non-anonymous) repository.
>
> 5. > Computational cost of converting a dense matrix to the butterfly structure
>
> Although this issue falls outside the claimed scope of the paper—which is to study Kronecker-sparse matrix multiplication on GPUs and to showcase its potential to accelerate the inference of models having Kronecker-sparse matrices (e.g., models trained from scratch with such matrices, or dense models replaced by Kronecker-sparse ones after training, with potential subsequent fine-tuning)—it is worth mentioning that approximating a given target matrix of size $m \times n$ by a product of Kronecker-sparse factors can be done efficiently in roughly $\mathcal{O}(mn)$ time [5].
>
> # References
> [1] Monarch: Expressive Structured Matrices for Efficient and Accurate Training, Dao et al, PMLR 2022.
>
> [2] Monarch mixer: A simple sub-quadratic GEMM-based architecture. Fu et al. NeurIPS, 2023.
>
> [3] Butterfly transform:An efficient FFT based neural architecture design. Vahid et al. CVPR, 2020.
>
> [4] Learning fast algorithms for linear transforms using butterfly factorizations. Dao et al. ICML, 2019
>
> [5] Butterfly factorization with error guarantees. Le et al., preprint, 2024.

---

> > ### Comment · Reviewer_nKiB · 2025-04-09
> >
> > Thank you for your kind reply. I have read your response, but I still believe that the conversion cost could be high when the proposed method operates on activations—such as in self-attention operations (e.g., Q @ Kᵀ, Attention-Score @ V)—which was my main concern. Since my doubts regarding this point have not been fully resolved, I would like to keep my score as it is.

---

### Official Review · Reviewer_nHJu · 2025-03-14

**Overall Recommendation:** 2

**Summary:**

This paper aims to speedup DNN inference with kronecker-sparse matrices by optimizing GPU memory accesses via customizing the CUDA kernels. The paper has made three key contributions: (1) analyzing the time and energy efficiency of existing implementations for multiplying kronecker-sparse matrices; (2) proposing a new tiling strategy in a new CUDA kernel implementation which reduces expensive GPU memory accesses; and (3) introducing a heuristic model that describes the time and energy efficiency. The experimental results have shown the proposed methods achieve 1.4x median speedup and 15% energy reduction.

**Claims And Evidence:**

While the paper has shown evidences that the proposed methods outperform existing frameworks, its evaluation results are not enough to fully support the claims. Specifically, the paper has never conducted ablation studies on how the percentage of non-zero elements affect the performance of the proposed kernels. This is important because when sparse matrix is used in practice, different level of sparsity might be used for better accuracy.

**Essential References Not Discussed:**

The paper should also discuss how their methods related to (sparse) tensor compilers, since these compilers can also be easily used to speedup sparse-kernel inference. Specifically, the paper should discuss why kronecker-sparse matrix multiplication is considered as a challenging problem and cannot just be considered as a special case for sparse tensor compilers:

* Ye, Zihao, et al. "Sparsetir: Composable abstractions for sparse compilation in deep learning." Proceedings of the 28th ACM International Conference on Architectural Support for Programming Languages and Operating Systems, Volume 3. 2023.
* Guan, Yue, et al. "Fractal: Joint multi-level sparse pattern tuning of accuracy and performance for DNN pruning." Proceedings of the 29th ACM International Conference on Architectural Support for Programming Languages and Operating Systems, Volume 3. 2024.

**Experimental Designs Or Analyses:**

Please refer to the "Methods and Evaluation Criteria" section of the review.

**Methods And Evaluation Criteria:**

The paper claims that they are able to support general inference of transformers. However, they only evaluate the vision transformers of small sizes, which could have very different trade-offs compared to transformer-based language models. Thus, it is obscure how the proposed method would perform in practice.

**Other Comments Or Suggestions:**

I feel the paper is in general well written. However, the section captions could be simplified as it makes more clear for the readers to understand the topic of each section.

**Other Strengths And Weaknesses:**

The motivation of the paper is also lacking. Specifically, the paper has never discussed how the kronecker-sparse matrix is used in practice. This is important because different sparsities levels could result in different trade-offs and design choices.

**Questions For Authors:**

* How do you measure the memory access time in general (see line 209, "We find that the memory rewritings can take up to 45% of the total runtime")
* How the proposed method speedup the inference of other types of networks such as GPT or Llama models?
* What is the cache hit ratio of reading/writing global memory of the proposed CUDA kernels?

**Relation To Broader Scientific Literature:**

While the paper has pointed out an interesting way to tackle the memory inefficiency in existing implementation of kronecker-sparse matrix multiplication, the paper should have discussed its relationship to other types of sparse matrix multiplication problems. In particular, similar issues have been seen in similar problems such as 3D sparse convolutions [1], where the permutations also needs to be done on inputs and outputs. The authors should have discussed the trade-offs of different design choices (e.g. input/output/weight stationary), and the reason why they pick up a specific one.

[1] Tang, Haotian, et al. "Torchsparse++: Efficient point cloud engine." Proceedings of the IEEE/CVF Conference on Computer Vision and Pattern Recognition. 2023.

**Theoretical Claims:**

I do not find any problems in the theoretical analysis in the paper but my expertise could be limited.

---

> ### Author Rebuttal · Authors · 2025-03-31
>
> Thank you for your review.
> # Regarding your questions
> 1. >How did we measure the time spent on memory rewritings
>
> We compared with the execution time where we removed the permutations/memory rewritings part, i.e. lines 1 and 3 in algorithm 1 (details in appendix B.2).
>
> 2. >Implications on other transformer sizes, e.g. larger transformers like GPT/Llama
>
> While we benchmark end-to-end latency on a transformer involving matrices of size 768 x 768, note that the benchmark covers sizes between 102 x 102 and 131072 x 131072, largely covering the wide range used in practice, even the ones in large transformers (e.g. 53248 x 53248 in Llama3-405B). Since the speedup of the kernel increases with the matrix size (section 4), our resource-limited evaluation on a small transformer actually corresponds to one of the most challenging setup to observe a speedup. The revision will make this explicit.
>
> 3. > Cache hit ratio
>
> We measured the cache hit as suggested by the reviewer and will add a discussion about it.
> # Regarding the other points you mentioned
> 4. >How Kronecker-sparse matrices are used in practice (motivation)
>
> Kronecker-sparse matrices are the building block of Butterfly matrices. The literature has proposed using them in a variety of way in neural networks (Table 1) and the benchmark considers patterns aligned with these different use cases. Therefore, the potential misalignment problem, where studied patterns would be different to the ones used in practice, does not arise here. The revision will make it clear.
>
> 5. >Relation to similar issues in sparse 3d convolutions
>
> A relevant analogy with this literature is that the new Algorithm 2 enables the implementation of a tiling strategy that optimises the dataflow in an analog way to how recent works [2,3,4] built on top of TorchSparse [1]. TorchSparse has three kernel calls: gather, multiplication, scatter. The subsequent works optimised this by overlapping memory and compute operations. For the sparse problem we consider, the dataflow optimisation enabled by Algorithm 2 has the same flavour: the kernel can now coalesce the input/output permutations with the multiplication part (Figure 3).
>
> 6. >Specify whether the kernel is input/output or weight-stationary
>
> The kernel is output-stationary, similarly to the dense cutlass kernels [5] and sparse 3d convolutional kernels [2,3,4] that have a similar dataflow. The revision will include a code template and explain this design choice as follows.
> * *Weight-stationary is not attractive*. In our Kronecker pattern $(a, b, c, d)$, when the parameter $a$ is large, the matrix is partitioned into $a$ submatrices that act on disjoint regions (Figure 2). Thus, weights are not reused across multiple input/output regions, limiting the benefits of keeping them stationary.
> * *Input-stationary is not attractive*. Due to the non-consecutive memory accesses involved (Figure 5), read and write operations on inputs/outputs come at a higher cost, so it is preferable to keep one of them stationary to reduce these costs. Both costs are largely driven by the parameter $d$ of the pattern $(a,b,c,d)$, which determines the distance between consecutive data elements that should be loaded together when considering one of the dense subproblems (Figure 5). Since their reuse costs are similar, we had to consider other factors. Input-stationarity poses parallelization challenges as different thread blocks cannot accumulate into the same output coefficient (no possible synchronization) [5]. In contrast, output-stationarity avoids this issue, hence our choice.
>
> 7. >Relation to sparse tensor compilers
>
> These compilers efficiently handle unstructured or simple block/tile sparsity, but Kronecker-sparse matrices feature a block-diagonal pattern with $b\times c$ sub-blocks further refined by $d\times d$ identity matrices. Capturing such nested sparsity would require combining an outer block-sparse format with an inner identity layout—something not supported off-the-shelf. Moreover, our Algorithm 2 leads to a tiling strategy with *non-contiguous* tiles along certain axes to reduce memory operations, a design choice that these compilers cannot accommodate.
>
> 8. >On adding results to show how the sparsity level affects speedups
>
> Thanks for the suggestion, the revision will include a graph showing how the speedup increases with the sparsity level (percentage of zero, ranging from 0 to 99% in the benchmark). The revision will also explain that the 12% of the 600+ patterns where the baselines still outperform the kernel correspond to cases with a large density of nonzero or a small value of the proposed heuristic, i.e., a small value of (b+c)/bc.
> # References
> [1] TorchSparse: Efficient Point Cloud Inference Engine. Tang et al 2022
>
> [2] Torchsparse++: Efficient point cloud engine. Tang et al CVPR 2023
>
> [3] SpConv, Yan 2022
>
> [4] SECOND: Sparsely Embedded Convolutional Detection. Yan et al 2018
>
> [5] github.com/NVIDIA/cutlass/blob/main/media/docs/efficient_gemm.md

---

### Official Review · Reviewer_z2VJ · 2025-03-15

**Overall Recommendation:** 2

**Summary:**

The paper "Fast inference with Kronecker-sparse matrices" focuses on optimizing matrix multiplication algorithms for Kronecker-sparse matrices, which are used in neural networks to reduce parameters while maintaining accuracy. The main contributions include:
- Benchmarking and Optimization: The authors benchmark existing GPU algorithms for multiplying Kronecker-sparse matrices and identify that up to 50% of runtime is spent on memory rewriting operations. They propose a new tiling strategy implemented in a CUDA kernel to reduce these memory transfers.
- New CUDA Kernel: The kernel achieves a median speedup of ×1.4 and reduces energy consumption by 15% compared to baseline implementations.
- Broader Impact: The new kernel accelerates the inference of neural networks, such as Vision Transformers (ViTs), by replacing dense layers with Kronecker-sparse matrices. This results in significant speedups in fully-connected layers.

Update after rebuttal:

I would like to sincerely thank the authors for taking the time to address all the issues I raised. However, as I pointed out in the initial review, the design and evaluation are quite limited in the single platform and software stacks. Even the authors claimed that they would provide the example in OpenCL, but it is hard for reviewers to evaluate without actually seeing the related implementation and experiments. So, for the current version, it may not meet the acceptance threshold for ICML.

**Claims And Evidence:**

The submission provides extensive evidence to support its claims, primarily through benchmarks and theoretical analyses. However, some aspects could be scrutinized for clarity and robustness:

Benchmarking Methodology: The paper presents a comprehensive benchmarking framework that compares various implementations of Kronecker-sparse matrix multiplication. The evidence is convincing, as it includes multiple scenarios and configurations, such as different memory layouts and precision levels (float and half-precision). However, the choice of specific hardware (NVIDIA GPUs) might limit the generalizability to other architectures.

Hardware Dependency: The performance benefits might not generalize equally across different hardware platforms.

Applicability to Other Architectures: The focus on ViTs leaves room for further research on how well the approach works with other neural network architectures.

Robustness Across Different Input Sizes: While the benchmark covers a range of sparsity patterns, the performance with very large or very small matrices might require additional validation.

**Essential References Not Discussed:**

No

**Experimental Designs Or Analyses:**

Hardware Dependency: While the paper focuses on NVIDIA GPUs, extending the benchmark to other hardware platforms could provide broader insights into the applicability of the proposed methods.

Generalizability Across Architectures: The paper primarily focuses on Vision Transformers. Investigating how well the new kernel performs with other neural network architectures (e.g., CNNs or RNNs) could enhance its utility.

Statistical Analysis: The benchmark results are presented using medians and interquartile ranges. While this provides a good overview, additional statistical analysis (e.g., hypothesis testing) might further validate the significance of the observed improvements.

Energy Consumption Measurements: The energy measurements are conducted on a different GPU (V100) than the time benchmarks (A100). While this is noted, ensuring consistency across all measurements could strengthen the conclusions regarding energy efficiency.

**Methods And Evaluation Criteria:**

The proposed methods and evaluation criteria in the paper on "Fast inference with Kronecker-sparse matrices" appear well-suited for the problem at hand, which is optimizing matrix multiplication algorithms for Kronecker-sparse matrices in neural networks.

The benchmarks are conducted on NVIDIA GPUs. Extending the evaluation to other hardware platforms, such as AMD GPUs or CPUs, could provide a more comprehensive understanding of the methods' applicability.

**Other Comments Or Suggestions:**

N/A

**Other Strengths And Weaknesses:**

Limited Hardware Scope: The paper focuses primarily on NVIDIA GPUs, which might limit the generalizability of the results to other hardware platforms like AMD GPUs or CPUs.

Originality of Heuristic: While the heuristic for efficient Kronecker-sparsity patterns is useful, it is based on a relatively straightforward analysis of memory operations. Further theoretical justification or exploration of its applicability beyond the current context could enhance its originality.

Clarity in Some Technical Details: Some sections, such as the explanation of perfect shuffle permutations, might be challenging for readers without a strong background in linear algebra. Additional explanations or references could improve clarity for a broader audience.

Broader Applicability: The paper primarily focuses on Vision Transformers. Exploring how the new kernel performs with other neural network architectures could further demonstrate its significance and versatility.

**Questions For Authors:**

1. How do you envision extending the new CUDA kernel to work efficiently on other hardware platforms, such as AMD GPUs or CPUs? Are there any specific challenges or opportunities you foresee in this process?

2. The heuristic based on the ratio (b+c)/bc is useful for identifying efficient Kronecker-sparsity patterns. Could you elaborate on how this heuristic might be refined or extended to accommodate different types of sparse matrices or computational contexts?

3. The paper highlights the impact of memory layout (batch-size-first vs. batch-size-last) on performance. How do you think this might influence the design of other neural network operations, and are there opportunities for further optimization in this area?

4. While the paper demonstrates significant speedups in Vision Transformers, what potential exists for applying these techniques to other neural network architectures, such as CNNs or RNNs? Are there specific challenges or opportunities in these contexts?

5. The new kernel not only improves time efficiency but also reduces energy consumption. Could you discuss how these energy savings might be further optimized or generalized across different hardware platforms?

6. How do you see Kronecker-sparse matrices comparing to other sparsity techniques, such as unstructured sparsity or other forms of structured sparsity? Are there scenarios where one might be preferred over the others?

**Relation To Broader Scientific Literature:**

The paper builds on the concept of butterfly matrices, which are structured matrices that can be expressed as products of sparse factors with specific sparsity patterns, often described by Kronecker products. Butterfly matrices have been used to accelerate linear transforms like the Discrete Fourier Transform (DFT) and the Hadamard Transform.

Prior work has shown that replacing dense matrices with sparse or structured matrices can improve neural network efficiency. For example, Dao et al. demonstrated that using butterfly matrices can speed up neural network training.

The paper introduces a new tiling strategy for matrix multiplication with Kronecker-sparse matrices, implemented in a CUDA kernel. This approach reduces memory transfers between different levels of GPU memory, leading to improved time and energy efficiency.

Similar heuristics have been used in other contexts to optimize sparse matrix operations, but the specific application to Kronecker-sparse matrices is novel and contributes to the broader literature on efficient neural network design.

**Theoretical Claims:**

Formal Proof for Algorithm Equivalence: While the paper explains the equivalence between Algorithm 1 and Algorithm 2, a formal proof might be beneficial for readers seeking rigorous mathematical validation.

Robustness of the Heuristic: The heuristic for efficient patterns is empirically validated but might benefit from further theoretical analysis to ensure its applicability across different scenarios or hardware platforms.

---

> ### Author Rebuttal · Authors · 2025-03-31
>
> Thank you for your review.
>
> # Regarding your questions
>
> 1. > Extending the kernel to other hardwares / Limited Hardware Scope
>
> We translated the CUDA kernel to openCL, so the kernel can now be used on other hardwares such as AMD GPUs or CPUs. The CUDA and openCL codes have also been integrated to an open-source python package available online. The revision will mention that and provide links to the (non-anonymous) repositories.
>
> 2. > Extending the heuristic beyond Kronecker-sparsity
>
> The heuristic is specifically tailored to Kronecker-sparse matrices with patterns (a, b, c, d), in order to predict the improvement in terms of memory transfer brought by the proposed kernel, compared to PyTorch baselines such as bmm. So there is no general and straightforward way to adapt it to other forms of sparsity and sparse multiplication algorithms. Nevertheless, counting the number of memory versus algebraic operations in a given sparse matrix multiplication for a certain sparsity pattern, and putting that in relation to its efficiency on a given hardware, is a general principle that might help in broader scenarios beyond Kronecker-sparse factors.
>
> 3. > Influence of batch-size-last versus batch-size-first on other neural network operations
>
> We agree that this is an interesting open question arising from the paper's results. For pointwise operations (e.g., activation functions like ReLU), changing the batch position is not expected to have any impact (and we indeed observed that internally). To investigate the impact of the batch position on other operations, it requires to carefully rewrite highly optimised kernels to convert them to batch-size-last, which requires time and expertise. Such a study falls out of the scope of what we claim to study (Kronecker linear layers), and is left to future work.
>
> 4. > Measuring speedups in other architectures, beyond Vision Transformers
>
> There is indeed an opportunity to extend the observations made in this paper to other architectures such as LLMs or RNNs, as they contain large linear layers where introducing Kronecker-sparsity is expected to speedup inference. The case of CNNs is more challenging. There are at least two ways to inject Kronecker sparsity in convolutional layers, none of which we found convincing enough to be considered in this paper. The first option—replacing the convolutional kernel K with a Kronecker-sparse matrix—is limited because of the small size of typical kernels (e.g., 7×7). The second option—recasting convolutions as matrix products with a butterfly-structured weight matrix as in [1]—requires in practice costly folding/unfolding operations on the inputs and outputs, making it impractical in our view.
>
> 5. > Generalisation of the speedups to other hardwares
>
> The claimed scope of the paper is to focus on NVIDIA GPUs as they are the most common hardware used in AI clusters. It is left open to explore the opportunities and challenges related to benchmarking and optimising the kernel on other hardwares. Since the paper comes with an openCL version of the kernel, people with specific hardware needs can now easily include it in their benchmark.
>
> 6. > Kronecker-sparsity versus other forms of sparsity
>
> Structured sparsity is clearly to be favored over unstructured sparsity for time and energy performance. Indeed, knowing the structure of the support in advance helps a lot to design efficient algorithms. Structured sparsity has also better theoretical proven properties (e.g., the functional space is closed and there always exists a minimizer of the loss for structured sparse networks, while it is not the case for unstructured ones, which might cause the solution to diverge, see theorem 4.2 in [2]). More specific comparisons would depend on the task, hardware, model and sparsity structure at hand.
>
> # Regarding the other points you mentioned
>
> 7. > Formal proof of the algorithm equivalence, background on permutation shuffles matrix
>
> Thank you for the suggestions, the revision will include both.
>
> 8. > Energy measurements on V100 while time measurement on A100
>
> The pyJoules package used in the paper to measure energy consumption is unfortunately not yet compatible with A100 GPUs.
>
> 9. > On the robustness of the benchmark for different matrix sizes
>
> We believe that the problem of robustness that you mention does not arise here, because the benchmark already covers all relevant matrix sizes encountered in practice. Indeed, it covers matrix sizes from 102×102 to 131072×131072 (6 orders of magnitude), while sizes typically used in transformers range from 500×500 to 15000 x 15000, and go up to 53248 in the largest models like Llama 3-405B [3].
>
> # References
>
> [1] Deformable Butterfly: A Highly Structured and Sparse Linear Transform. Lin et al NIPS 2022
>
> [2] Does a sparse ReLU network training problem always admit an optimum? Le et al NIPS 2023.
>
> [3] The Llama 3 Herd of Models, 2024.

---

> > ### Comment · Reviewer_z2VJ · 2025-04-05
> >
> > I would like to sincerely thank the authors for taking the time to address all the issues I raised. However, as I pointed out in the initial review, the design and evaluation are quite limited in the single platform and software stacks. Even the authors claimed that they would provide the example in OpenCL, but it is hard for reviewers to evaluate without actually seeing the related implementation and experiments. So, for the current version, it may not meet the acceptance threshold for ICML.

---

### Decision · Program_Chairs · 2025-05-01

**Decision:**

Accept (poster)

**Comment:**

This paper introduces a specialized CUDA kernel for accelerating Kronecker-sparse matrix multiplications, a structured form of sparsity increasingly used in neural networks. The authors identify a key inefficiency in existing implementations — excessive memory rewriting — and propose a new tiling strategy to reduce this overhead. The result is a median 1.4× speedup and 15% energy savings, validated across over 600 sparsity patterns and demonstrated in accelerating inference for Vision Transformers (ViTs) on NVIDIA GPUs. Through the rebuttal, the authors provided detailed clarifications and extended their implementation to OpenCL to address generalization concerns.

Overall, the paper makes a technically sound and practically useful contribution to structured sparsity and efficient inference. It is well-executed, timely, and relevant. While some concerns remain about the generality and integration with broader toolchains, the core idea is novel and impactful.

Given the strength of the engineering contribution, the quality of the benchmarks, and the clarity of the rebuttal, this paper meets the bar for acceptance. Minor revision would further strengthen the work for publication.